# LEARNING ADMISSIBLE HEURISTICS FOR A*: THEORY AND PRACTICE

**Ehsan Futuhi[1] , Nathan R. Sturtevant[1,2]**
[1]Department of Computing Science, University of Alberta    [2]Alberta Machine Intelligence Institute (Amii)
`{futuhi, nathanst}@ualberta.ca`

## ABSTRACT

Heuristic functions are central to the performance of search algorithms such as A*, where *admissibility*—the property of never overestimating the true shortest-path cost—guarantees solution optimality. Recent deep learning approaches often disregard admissibility and provide limited guarantees on generalization beyond the training data. This paper address both of these limitations. First, we pose heuristic learning as a constrained optimization problem and introduce *Cross-Entropy Admissibility (CEA)*, a loss function that enforces admissibility during training. On the Rubik's Cube domain, this method yields near-admissible heuristics with significantly stronger guidance than compressed pattern database (PDB) heuristics. Theoretically, study the sample complexity of learning heuristics. By leveraging PDB abstractions and the structural properties of graphs such as the Rubik's Cube, we tighten the bound on the number of training samples needed for A* to generalize. Replacing a general hypothesis class with a ReLU neural network gives bounds that depend primarily on the network's width and depth, rather than on graph size. Using the same network, we also provide the first generalization guarantees for *goal-dependent* heuristics.

## 1 INTRODUCTION

Heuristic search algorithms such as A* (Hart et al., 1968) are widely used in pathfinding tasks (Yonetani et al., 2021; Meng et al., 2024; Kirilenko et al., 2023), where the objective is to find a least-cost path from a start state to a goal state in a graph. These algorithms rely on a *heuristic function* that estimates the cost-to-go from a state to the goal. To guarantee optimality, A* requires the heuristic to be *admissible*, meaning it must never overestimate the true cost, $h^*$, to the goal. If we wish to integrate machine learning with heuristic search algorithms like A* we have several possible approaches: (1) We can adapt learning to meet the requirements of search algorithms (Samadi et al., 2008; Li et al., 2022), (2) we can adapt search algorithms to meet the properties of ML heuristics (Lelis et al., 2021), (3) we can do a combination of both, or (4) we can abandon all theoretical guarantees, hoping for high quality solutions or fast search in practice (Orseau et al., 2023; Hazra et al., 2024; Chen et al., 2025; Huang et al., 2025). This paper studies the foundations of the first question: what is the sample complexity needed to learn admissible heuristics, what is an effective loss function for training, and how close we can get to optimal heuristics in practice?

Traditionally, admissible heuristic functions are computed from domain knowledge of the problem (Katz & Keyder, 2022; Haslum et al., 2005; Seipp, 2024). One prominent example is pattern databases (PDBs) (Korf, 1997; Culberson & Schaeffer, 1998). PDBs result from abstracting the problem and storing $h^*$ for all states in the abstract problem. Learning heuristics from data (Li et al., 2022; Agostinelli et al., 2021; Pándy et al., 2022; Shen et al., 2020; Chen et al., 2024b) has become increasingly popular for two main reasons: (1) designing good heuristics for complex environments is challenging, and (2) data-driven deep learning algorithms have demonstrated outstanding performance in analogous tasks (Silver et al., 2017; Schrittwieser et al., 2020; Wurman et al., 2022; Touvron et al., 2023). While learned heuristic functions may outperform traditional heuristics, they typically lose admissibility and thus suboptimality guarantees (Agostinelli et al., 2019; Yonetani et al., 2021; Archetti et al., 2021; Kirilenko et al., 2023). Despite substantial empirical progress, the theoretical foundations of heuristic learning are relatively underexplored. A central question is: *How many training samples are required to ensure that the learned heuristic generalizes effectively to the entire*

*graph?* Sakaue & Oki (2022) addressed this question for Greedy Best-First Search (GBFS) and A*, providing upper and lower bounds on the training samples necessary to ensure generalization.

We extend this work to the problem of learning admissible heuristics in two main directions. First, we analyze the sample complexity of learning heuristic functions from a given dataset. We introduce a new upper bound for the expected suboptimality of A* with reopenings. We then derive tighter generalization bounds by leveraging structural properties of the graphs of interest. We further incorporate neural networks as the heuristic model and derive generalization bounds that depend primarily on the size of the network rather than the size of the underlying graph. Notably, using neural networks, we establish the first generalization bounds for goal-dependent heuristics. Across all theoretical results, we demonstrate that leveraging PDBs instead of drawing training samples from the original graph leads to improved bounds.

Second, we formulate the optimization problem for learning admissible heuristics for search applications. We propose a novel loss function, termed *Cross Entropy Admissibility (CEA)*, along with a training framework that satisfies all the constraints of the optimization problem. We evaluate our framework on several $3 \times 3$ Rubik's Cube pattern databases PDBs. The CEA loss function produces near-admissible heuristics, achieving inadmissibility rates around $1 \times 10^{-6}$ across all PDBs—significantly outperforming standard training based on the *Cross Entropy (CE)* loss. The strength of the learned heuristics exceeds that of same-sized PDBs obtained using classical compression techniques; notably, for the 8-corner PDB, the CEA loss successfully learns the admissible PDB heuristic perfectly.

## 2 BACKGROUND

In heuristic search, the task is to find a path from a start state $s_{init} \in V$ to a goal state $goal \in V$ in a graph $\{G = \{V, E\}, c, h\}$, where $c \colon E \to \mathbb{R}^+$ specifies the cost for each edge and the heuristic function $h(v)$ estimates the distance from any state $v$ to $goal$. A heuristic is called *admissible* if, for all $v \in V$, we have $h(v) \leq h^*(v)$, where $h^*(v)$ is the true shortest distance from $v$ to $goal$. A heuristic function $h$ is *consistent* if, for all states $a, b \in V$ such that $(a, b) \in E$, we have $h(a) \leq c(a, b) + h(b)$. In large state spaces, the graph $G$ is represented implicitly and is generated dynamically during search by expanding states and exploring their neighbors. Although A* guarantees optimal solutions under an admissible heuristic, it can become less efficient if the heuristic is inconsistent, due to the potential for re-expanding nodes (Martelli, 1977; Felner et al., 2011; Helmert et al., 2019). For a positive integer $d$, write $[d] \coloneqq \{1, 2, \ldots, d\}$. Given vectors $\{\mathbf{x}^1, \ldots, \mathbf{x}^t\} \subseteq \mathbb{R}^d$, we use superscripts to index the vectors and subscripts for coordinates, so $\mathbf{x}_j^i$ denotes coordinate $j$ of vector $i$. For any real vector $\mathbf{x}$, the operator $\lceil \mathbf{x} \rceil$ applies the ceil function *coordinatewise*. We use $\mathbb{I}(\cdot)$ to denote the indicator function, which returns 1 when the stated condition holds and 0 otherwise. Let $\Pi$ denote the space of problem instances and let $D$ be an unknown distribution supported on $\Pi$; we write $x \sim D$ for a random instance. Each $x \in \Pi$ specifies a start state $s_{init}$ from which the search is initiated. We make the following assumption on all instances.

**Assumption 1** (Fixed graph and reachability). *The state graph $G$ and $goal \in V$ are fixed and shared across all instances $x \in \Pi$. For every instance with start state $s_{init} \neq goal$, there exists at least one directed path from $s_{init}$ to goal.*

**A\* Algorithm.** We use $A_h^*$ to denote an A* algorithm that employs a heuristic function $h$. This algorithm maintains two lists of states: OPEN for states that have been generated but not expanded, and CLOSED for states that have already been expanded. Additionally, we store a pointer to the parent of state $s$, denoted as Parent($s$), which allows us to reconstruct the path to $s_{init}$ by tracing backward from $s$. Algorithm 1 in Appendix B provides an overview of the procedure of $A_h^*$. Given an instance $x \in \Pi$, $A_h^*$ initializes its search by adding $s_{init}$ to OPEN. At each iteration, it selects from OPEN a state $s$ with the minimum value of $f(s) = h(s) + g(s)$. To ensure consistent performance bounds for $A_h^*$, we impose the following assumption when ties occur.

**Assumption 2.** *If the state space $V$ contains $n$ vertices, we impose an arbitrary strict total order on $V$. For example, consider the order $v_1 < v_2 < \cdots < v_n$. In this ordering, if two states $v_i$ and $v_j$ have the same lowest $f$ value, we select $v_i$ when $i < j$, and $v_j$ otherwise.*

When expanding a state $s$, let $g_{\text{new}} \leftarrow g(s) + c(s, s')$ denote the generated cost of the path from the start through $s$ to its child $s'$. Three situations can arise for $s'$ based on $g_{\text{new}}$:

1. If $s' \notin$ OPEN $\cup$ CLOSED, then we add $s'$ to OPEN.

2. If $s' \in$ OPEN and $g_{\text{new}} < g(s')$, then we update Parent$(s')$ and $g(s')$ to reflect the improved path.

3. If $s' \in$ CLOSED and $g_{\text{new}} < g(s')$, then we update Parent$(s')$, $g(s')$, and move $s'$ from CLOSED back to OPEN.

Case 3 is only needed if the heuristic is inconsistent. After generating and evaluating all children of $s$, we remove $s$ from OPEN and place it in CLOSED. This procedure continues until the *goal* state is selected from OPEN for expansion, at which point the algorithm terminates with a path to *goal*.

**Abstraction-Based Heuristics.** As a form of abstraction heuristics, Pattern Databases (PDBs) simplify a graph $V$ by applying a homomorphic transformation to obtain a reduced state space $\phi(V)$. For example, a $3 \times 3$ Rubik's Cube contains 26 cubies—12 edges, 8 corners, and 6 centers—as illustrated in Figure 1. One can abstract away the edges and centers to construct an 8-corner PDB with $8! \times 3^7$ states, in contrast to the $4.33 \times 10^{19}$ states in the original graph. Within this abstract representation, if there is an edge between $v_1$ and $v_2$ in $V$, an edge will also be present between $\phi(v_1)$ and $\phi(v_2)$ in $\phi(V)$.

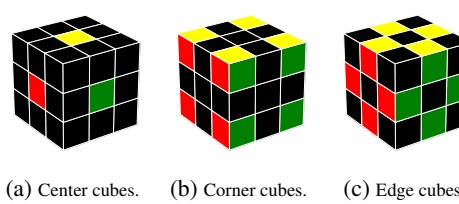

(a) Center cubes.    (b) Corner cubes.    (c) Edge cubes.

Figure 1: A solved $3 \times 3$ Rubik's Cube with each group of cubies shown separately: (a) center cubies, (b) corner cubies, and (c) edge cubies.

Thus, all distances are admissible, and all states from $V$ that reduce to the same state in $\phi(V)$ have identical heuristic values. More details regarding the reduction from states in $V$ to states in $\phi(V)$ are shown in Figure 3 in Appendix C. The PDB stores the perfect heuristic value (optimal distance to the abstract goal) for every abstract state, obtained by running an optimal search in the abstract state space. During the search, the current state is mapped to $\phi(v)$, and a ranking function (Myrvold & Ruskey, 2001) is used to assign a unique integer to the abstract state. This functions as an index in the lookup table to retrieve the the precomputed distance to the abstract goal, which serves as the heuristic value $h(v)$. Compression techniques (Felner et al., 2007; Helmert et al., 2017) reduce the size of the PDB by grouping entries and only storing the lowest heuristic value from the group, which maintains admissibility.

**Preliminaries for Sample Complexity Analysis.** We use *pseudo-dimension* (Pollard, 1984), a central concept for measuring the complexity of a class of real-valued functions, as the foundation of our analysis. The definition of *pseudo-dimension* is as follows.

**Definition 1.** *Let $\mathcal{H} \subseteq \mathbb{R}^{\mathcal{Y}}$ be a set of functions mapping a domain $\mathcal{Y}$ into $\mathbb{R}$. We say that a subset $\{y_1, \ldots, y_N\} \subseteq \mathcal{Y}$ is shattered by $\mathcal{H}$ if there exist target values $z_1, \ldots, z_N \in \mathbb{R}$ such that*

$$\left| \left\{ \left( \mathbb{I}\{h(y_1) \geq z_1\}, \ldots, \mathbb{I}\{h(y_N) \geq z_N\} \right) \mid h \in \mathcal{H} \right\} \right| = 2^N.$$

*The largest number of samples that can be shattered by $\mathcal{H}$ is called the* pseudo-dimension *of $\mathcal{H}$, denoted by* $\mathrm{Pdim}(\mathcal{H})$.

We use the following proposition (Theorem 11.8 in (Mohri et al., 2018)) that helps us to attain generalization bounds using the definition of pseudo-dimension.

**Proposition 1.** *Let $H > 0$, $\mathcal{H} \subseteq [0, H]^{\mathcal{Y}}$, and $\mathcal{D}$ be a probability distribution over $\mathcal{Y}$. Suppose we draw $\{y_1, \ldots, y_N\} \sim \mathcal{D}^N$ i.i.d. Then, with probability at least $1 - \delta$ over this random draw, the following holds for all $h \in \mathcal{H}$:*

$$\left| \frac{1}{N} \sum_{i=1}^{N} h(y_i) - \mathbb{E}_{y \sim \mathcal{D}} \left[ h(y) \right] \right| = \mathrm{O}\left( H \sqrt{\frac{\mathrm{Pdim}(\mathcal{H}) \log\left(\frac{N}{\mathrm{Pdim}(\mathcal{H})}\right) + \log\left(\frac{1}{\delta}\right)}{N}} \right).$$

In simpler terms, if the size of the training dataset is $N = \Omega\left( \frac{H^2}{\epsilon^2} \left( \mathrm{Pdim}(\mathcal{H}) \log \frac{H}{\epsilon} + \log \frac{1}{\delta} \right) \right)$, with a probability of $1 - \delta$, we can guarantee that the difference between true expectation over the entire graph and the emperical mean over the dataset is less than $\epsilon$.

**Performance measure.** We need a general definition of performance that can cover various metrics. Specifically, we measure the performance of $A_h^*$ on $x \in \Pi$ using a utility function $u$, defined below:

**Assumption 3** (Sakaue & Oki (2022, Assumption 3)). *Let $H > 0$. A utility function $u$ takes $x$ and a series of all* OPEN, CLOSED, *and* Parent$(\cdot)$ *generated during the execution of $A_h^*$ on $x \in \Pi$ as input, and returns a scalar value in $[0, H]$.*

With this definition, the utility function could, for instance, be the number of node expansions or the total running time, since both can be directly computed from the sizes of the OPEN and CLOSED lists. As shown in Proposition 1, we require a strict upper bound on these measures to ensure that all functions in the class $\mathcal{U}$ return positive, bounded values. This assumption is explicitly used in experiments, as we often set time or memory constraints when running $A_h^*$. We denote by $u_h : \Pi \to [0, H]$ the utility function that evaluates $A_h^*$'s performance on any instance $x \in \Pi$, and we define the class of such functions as $\mathcal{U} = \{ u_h : \Pi \to [0, H] \mid h \in \mathbb{R}^n \}$.

Our goal is to determine the number of training instances needed to learn a heuristic function $h \in \mathbb{R}^n$ such that the $A_h^*$ algorithm achieves optimal performance w.r.t the utility function on all instances $x \sim \mathcal{D}$. Concretely, we want to learn a heuristic function that attains optimal expected performance, $\mathbb{E}_{x \sim \mathcal{D}}[u_{\hat{h}}(x)]$. Because we only observe the performance of $A_h^*$ on the training instances $x_1, \ldots, x_N$, namely $u_h(x_1), \ldots, u_h(x_N)$, we must ensure generalization to any $x \sim \mathcal{D}$. To achieve this, we need to bound the difference between the empirical average performance and the expected performance,

$$\left| \frac{1}{N} \sum_{i=1}^{N} u_h(x_i) \ - \ \mathbb{E}_{x \sim \mathcal{D}}[u_h(x)] \right|$$

uniformly for all $h \in \mathbb{R}^n$. Establishing such a bound requires determining the pseudo-dimension of $\mathcal{U}$. In the following section, we use our theoretical tools to derive this result. Note that all proofs and an extended related work discussion are provided in the appendix.

## 3 SAMPLE COMPLEXITY ANALYSIS

In this section, we attain the $\mathrm{Pdim}(\mathcal{U})$ for the $A_h^*$ algorithm. To this aim, we need to discretize $\mathbb{R}^n$ into regions such that all heuristics in a region have the same performance: $u_{h_1} = u_{h_2}$ if $h_1, h_2 \in \mathcal{P}$. If we look at the Algorithm 1, the only part that the heuristic function impacts the performance is the node selection step from OPEN (line 3). However, in addition to $h$, the path cost from $s_{init}$ to each state $v$ also matters to determine the total cost: $f(v) = g(v) + h(v)$. If only $h$ determines which node to expand, such as in GBFS, we know that only the total ranking given by $h$ over all states $v \in V$ matters. So, the number of regions are the total number of different ordering among the states, which is $n!$ (Sakaue & Oki, 2022; Chrestien et al., 2024). In the case of A* algorithm, we first need to define the cases when two different heuristics result in the same performance:

**Lemma 1.** *Let $h$ and $h'$ be two heuristic functions in $\mathbb{R}^n$ such that $h(v_i) - h(v_j) = h'(v_i) - h'(v_j)$ for any two pairs of $v_i, v_j \in V$. Then, it follows that $u_h(x) = u_{h'}(x)$ for every $x \in \Pi$.*

Before presenting our main theorem, we require a more detailed examination of $g$-costs, which appear as one component of the $f$-cost. Formally, at each iteration of A*, $g(v)$ tracks the cost of the shortest path discovered so far from $s_{init}$ to a vertex $v$. This cost may be updated if a lower-cost path is found later in the search, implying that $g(v)$ always represents an upper bound on the true cost from $s_{init}$ to $v$. To determine the number of distinct $g(v)$ values in a graph, we impose the following assumption, which holds for many combinatorial problems (e.g., the Rubik's Cube).

**Assumption 4.** *Let $\{G = \{V, E\}, c, h\}$ define our task. The edge cost function $c \colon E \to \{c_0\}$ assigns a constant cost $c_0$ to all edges.*

Under Assumption 4, the number of distinct values that $g(v)$ can take for any $v \in V$ is at most $|V| = n$. Define $\mathcal{G}_V = \{ (v, g(v)) \mid v \in V \}$, which is the set of vertex–cost pairs. Hence, $|\mathcal{G}_V| \leq n^2$. Armed with these observations, we now establish an upper bound on $\mathrm{Pdim}(\mathcal{U})$ for A* in the following theorem. While our proof uses a different derivation technique, the resulting bound matches that of Sakaue & Oki (2022).

**Theorem 1.** *Let $A_h$ have parameters $h \in \mathbb{R}^n$, and let the graph $G$ satisfy Assumption 4. Then, it holds that $\mathrm{Pdim}(\mathcal{U}) = \mathcal{O}(n \log n)$.*

**Extension to PDB Heuristics.** In general, we consider heuristic functions $h \in \mathbb{R}^n$. However, for PDB heuristics, one can learn values on an abstracted state space whose size is $m$, which can be exponentially smaller than $n$. In that case, $h$ effectively lives in $\mathbb{R}^m$. We next show that learning heuristic functions to approximate a PDB heuristic can further reduce $\mathrm{Pdim}(\mathcal{U})$.

**Theorem 2.** *Let $G$ be a graph with $|G| = n$, and let $P$ be a PDB dataset over $G$ with an induced graph of size $m$. Suppose $A_h$ uses a heuristic $h$ trained on $P$ and that $G$ satisfies Assumption 4. Then, $\mathrm{Pdim}(\mathcal{U}) = \mathcal{O}(m \log n)$.*

**Remarks on upper bounds for $\mathbf{Pdim}(\mathcal{U})$.** Sakaue & Oki (2022) provided the first upper bounds on $\mathrm{Pdim}(\mathcal{U})$. For a general class of graphs, they proved an upper bound of $\mathcal{O}(n^2 \log n)$, later refining it to $\mathcal{O}(n \log(nW))$ under the assumption that edge weights are bounded by a constant $W$. In our setting, where graphs follow Assumption 4, we initially obtain an upper bound of $\mathcal{O}(n \log n)$. We then improve this bound to $\mathcal{O}(m \log n)$ by utilizing learned PDB heuristics, with $m$ representing the size of the graph induced by the PDB.

### 3.1 Upper bounds on the expected suboptimality

If we aim to achieve $\mathrm{Pdim}(\mathcal{U})$ for the A* algorithm *without* imposing Assumption 4, the sample complexity scales on the order of $\mathcal{O}(n^2 \log n)$. Consequently, to obtain tighter results, one often restricts the class of graphs under consideration. The general class of performance measures defined earlier encompasses a wide range of metrics related to the performance of the $A_h^*$ algorithm, including runtime, the number of node expansions, and memory consumption. Rather than seeking tighter bounds for this general class, we focus on a special measure called the *expected suboptimality*, which captures how far the solution returned by A* is from the optimal solution. For a given problem instance $x \in \Pi$, let $C(x)$ be the cost of the path found by A*, and let $C^*(x)$ be the cost of an optimal path. We define the suboptimality as

$$u_h(x) = C_h(x) - C^*(x).$$

Then, applying the bound on $\mathrm{Pdim}(\mathcal{U})$ together with Proposition 1, we obtain the following upper bound on the gap between the *expected* and *empirical* suboptimality on training data:

$$\mathbb{E}_{x \sim \mathcal{D}}\big[C_h(x) - C^*(x)\big] \leq \frac{1}{N} \sum_{i=1}^{N}(C_h(x_i) - C^*(x_i)) + \widetilde{\mathcal{O}}\Big(H\sqrt{\frac{n^2 + \log\frac{1}{\delta}}{N}}\Big). \quad (1)$$

To reduce the error term in Equation (1), we employ a worst-case bound on the quality of the solution returned by A*. Valenzano et al. (2014) proved that if A* does not reopen nodes (i.e., if we remove lines 13–14 in Algorithm 1), then the suboptimality can be bounded by the total *inconsistency* along the edges of an optimal path. Importantly, this bound holds for any heuristic function—even if it is not admissible—as long as the following conditions are satisfied for all $s \in V$:

- $h(s) \geq 0$,
- $h(s) = 0$ if $s = goal$, and
- $h(s) \neq \infty$ whenever $h^*(s) \neq \infty$.

Concretely, for any instance $x \in \Pi$, if $v_0, v_1, \ldots, v_k$ is an optimal path with cost $C^*(x)$, then

$$C_h(x) - C^*(x) \leq \sum_{j=1}^{k-1} \mathrm{INC}_h\big(v_j, v_{j+1}\big),$$

where, for every edge $(p, c) \in E$, the *inconsistency* $\mathrm{INC}_h$ is defined as $\mathrm{INC}_h(p, c) = \max\big\{ h(p) - h(c) - c(p, c), 0\big\}$. Because $\mathrm{INC}_h(\cdot, \cdot)$ is always non-negative, the total inconsistency on any path can be estimated by summing the edge inconsistencies along that path. Sakaue & Oki (2022) further showed that this bound remains valid even when A* reopen nodes from CLOSED upon discovering a path with lower $g$ cost. In Theorem 3, we show that the bound can be tightened when A* is allowed to reopen nodes.

**Theorem 3.** *Let $x \in \Pi$ be a problem instance, and let $P_{\mathrm{opt}} = v_0, v_1, \dots, v_k$ be its optimal solution path with cost $C^*(x)$. Suppose A\* is allowed to reopen nodes. Then the cost $C_h(x)$ of any solution returned by A\* satisfies*

$$C_h(x) \; - \; C^*(x) \;\; \leq \;\; \max_{v \in P_{\mathrm{opt}}} \Big[ h(v) \; - \; h^*(v) \Big].$$

We use $\Psi_h(x) = \max_{v \in P_{\mathrm{opt}}} \big[ h(v) - h^*(v) \big]$ to represent the inadmissibility function parameterized by $h$ for $x$. We define the class of all inadmissibility functions as $\hat{\mathcal{U}} = \{ \, \Psi_h : \Pi \to [0, \hat{H}] \; \big| \; h \in \mathbb{R}^n \}$, where $\hat{H}$ is the bound on the performance measure required by Assumption 3. In the following theorem, we bound the difference between the empirical inadmissibility over $N$ training instances, namely $\frac{1}{N} \sum_{i=1}^{N} \Psi_h(x_i)$, and the expected inadmissibility using $\mathrm{Pdim}(\hat{\mathcal{U}})$.

**Theorem 4.** *For the class $\hat{\mathcal{U}}$ of inadmissibility functions, it holds that $\mathrm{Pdim}(\hat{\mathcal{U}}) = \mathrm{O}(n \log n)$ and we can bound the generalization error by*

$$\mathop{\mathbb{E}}_{x \sim \mathcal{D}} \big[ \Psi_h(x) \big] \;\; \leq \;\; \frac{1}{N} \sum_{i=1}^{N} \Psi_h(x_i) \; + \; \tilde{\mathrm{O}} \Big( \hat{H} \sqrt{\tfrac{n + \log \frac{1}{\delta}}{N}} \Big). \tag{2}$$

Now, looking at the generalization error from Theorem 4 and Theorem 3, we get a tighter upper bound for the expected suboptimality:

$$\mathop{\mathbb{E}}_{x \sim \mathcal{D}} [C_h(x) \; - \; C^*(x)] \leq \mathop{\mathbb{E}}_{x \sim \mathcal{D}} [\Psi_h(x)] \leq \frac{1}{N} \sum_{i=1}^{N} \Psi_h(x_i) + \tilde{\mathrm{O}} \left( \hat{H} \sqrt{\frac{n + \log \frac{1}{\delta}}{N}} \right). \tag{3}$$

## 3.2 Sample Complexity using Neural Networks

So far, we have analyzed the sample complexity of heuristic functions $h \in \mathbb{R}^n$ under the assumption that each vertex $v \in V$ has a distinct value, and modifying the heuristic at one vertex does not affect the others. However, in many data-driven settings, these heuristics are realized by a neural network rather than a simple mapping in $\mathbb{R}^n$. In this section, we derive sample complexity bounds for neural network–based heuristic functions on graphs that satisfy Assumption 4. To do so, we first formalize both the mapping induced by these networks and our corresponding hypothesis class.

**Definition 2** (Neural networks Cheng et al., Definition 2.4). *Given any function $\sigma : \mathbb{R} \to \mathbb{R}$, we will use the notation $\sigma(\mathbf{x})$ for $\mathbf{x} \in \mathbb{R}^d$ to mean $[\sigma(\mathbf{x}_1), \sigma(\mathbf{x}_2), \dots, \sigma(\mathbf{x}_d)]^{\mathsf{T}} \in \mathbb{R}^d$. Let $\sigma : \mathbb{R} \to \mathbb{R}$ and let $L$ be a positive integer. A neural network with* activation $\sigma$ and architecture *$\boldsymbol{w} = [w_0, w_1, \dots, w_L, w_{L+1}]^{\mathsf{T}} \in \mathbb{Z}_{+}^{L+2}$ is a parameterized function class, parameterized by $L+1$ affine transformations $\{T_i : \mathbb{R}^{w_{i-1}} \to \mathbb{R}^{w_i}, i \in [L+1]\}$ with $T_{L+1}$ linear, is defined as the function*

$$T_{L+1} \circ \sigma \circ T_L \circ \cdots T_2 \circ \sigma \circ T_1.$$

*$L$ denotes the number of hidden layers in the network, while $w_i$ signifies the width of the $i$-th hidden layer for $i \in [L]$. The input and output dimensions of the neural network are denoted by $w_0$ and $w_{L+1}$, respectively. If $T_i$ is represented by the matrix $A^i \in \mathbb{R}^{w_i \times w_{i-1}}$ and vector $\mathbf{b}^i \in \mathbb{R}^{w_i}$, i.e., $T_i(\mathbf{x}) = A^i \mathbf{x} + \mathbf{b}^i$ for $i \in [L+1]$, then the weights of neuron $j \in [w_i]$ in the $i$-th hidden layer come from the entries of the $j$-th row of $A^i$ while the bias of the neuron is indicated by the $j$-th coordinate of $\mathbf{b}^i$. The size of the neural network is defined as $w_1 + \cdots + w_L$, denoted by $U$.*

With reference to Definition 2, a family of neural networks is defined as $N^{\sigma} : \mathbb{R}^{w_0} \times \mathbb{R}^W \to \mathbb{R}^{w_{L+1}}$, where $\mathbb{R}^{w_0}$ is the input space, $\sigma$ is the activation function in hidden layers, and $\mathbb{R}^W$ is the parameter space. The parameter space $\mathbb{R}^W$ is implicitly defined by all weight matrices $A^i$ and bias vectors $\mathbf{b}^i$ for $i \in [L+1]$ within the neural network. Each function $N^{\sigma}(\mathbf{x}, \mathbf{w})$ is defined for any $\mathbf{x} \in \mathbb{R}^{w_0}$ and $\mathbf{w} \in \mathbb{R}^W$ as

$$N^{\sigma}(\mathbf{x}, \mathbf{w}) \; = \; T_{L+1}\big( \sigma\big(T_L(\dots T_2(\sigma(T_1(\mathbf{x}))) \dots )\big)\big),$$

where each $T_i$ is an affine transformation that depends on $\mathbf{w}$. For the hidden layers, we employ the **ReLU** activation, while the last layer uses the **Softmax** activation, consistent with treating heuristic learning as a classification task.

- **ReLU:** The Rectified Linear Unit (ReLU) activation $\text{ReLU} : \mathbb{R} \to \mathbb{R}_{\geq 0}$ is given by $\text{ReLU}(x) = \max\{0, x\}$.

- **Softmax:** The $\text{Softmax}$ activation $: \mathbb{R}^\ell \to (0, 1)^\ell$ is applied coordinatewise:

$$\text{Softmax}(x_i) \;=\; \frac{\exp(x_i)}{\sum_{j=1}^{k} \exp(x_j)}, \quad i = 1, \ldots, \ell.$$

The parameterization induced by the neural network $N^{\sigma, \sigma'}$ is shown as $\varphi_{\mathbf{w}}^{N^{\sigma, \sigma'}}$, where $\sigma$ is the hidden-layer activation function and $\sigma'$ is the activation function in the final layer. This gives rise to the performance class $\mathcal{U} = \{ u_h : \Pi \to [0, H] \mid h = \varphi_{\mathbf{w}}^{N^{\sigma, \sigma'}}, \mathbf{w} \in \mathbb{R}^W \}$. In the following theorem, we derive $\text{Pdim}(\mathcal{U})$.

**Theorem 5.** *Let $G$ be a graph with $|V| = n$, and let $P$ be a PDB dataset over $G$ whose induced graph has size $m$. Assume $G$ satisfies Assumption 4, and let $h : V \to [0, D]$ be a heuristic function trained via a neural network. Then, $\text{Pdim}(\mathcal{U}) = \mathcal{O}(n)$. Moreover, if $h$ is trained on $P$, it follows that $\text{Pdim}(\mathcal{U}) = \mathcal{O}(m)$.*

The bound in Theorem 5 depends on the size of the graph, which may be large. Ideally, we want to leverage neural networks to obtain a tighter bound in terms of the network's size. The proposed approach from Balcan et al. (2021a) is not guaranteed to remain valid under a neural network's parameterization. So, we first define a representation function and then derive a bound on $\text{Pdim}(\mathcal{U})$ that accounts for the parameters of the neural network.

**Definition 3.** *For each instance $I \in \mathcal{I}$, we define a representation function $Rep(I) = \mathbf{x}$, where $\mathbf{x}$ is the feature vector fed into the neural network. This representation encompasses all relevant features of an instance, including any common across instances.*

Our definition of $Rep(I)$ must encompass all the information required to find a path from the start state to the goal. However, we cannot simply include all states in each instance, since computing $h^*$ for all states is itself the ultimate goal of training. Instead, for each instance, we include a set $B$ of states encountered by running a search from the start state for that instance to the goal using $h^*$. This approach allows us to capture only the necessary data without relying on all possible states.

**Theorem 6.** *Let $G$ be a graph with $|V| = n$, and let $P$ be a PDB dataset over $G$ whose induced graph has size $m$. Assume $G$ satisfies Assumption 4, and let $h : V \to [0, D]$ be a heuristic function trained via a neural network. Then, $\text{Pdim}(\mathcal{U}) = \mathcal{O}\left(LW \log(U + \ell) + W \log(\ell|B|(L + 1))\right)$. Moreover, if $h$ is trained on $P$, it follows that $\text{Pdim}(\mathcal{U}) = \mathcal{O}\left(LW \log(U + \ell') + W \log(\ell'|B'|(L + 1))\right)$.*

**Instance-dependent framework.** So far, our analysis of sample complexity has focused on heuristic functions that can be used for all instances, under the assumption that each instance shares the same goal state, and thus the same heuristic values. However, we can also view instances as $s_{init}-goal$ pairs, where changing the goal requires a different heuristic. To address this, we adopt neural networks as the learning framework, enabling heuristic values that adapt to instance-specific features.

**Theorem 7.** *Let $G$ be a graph with $|V| = n$, and let $P$ be a PDB dataset over $G$ whose induced graph has size $m$. Assume that $G$ satisfies Assumption 4, and let $h : V \to [0, D]$ be a heuristic function trained via a neural network, capable of estimating heuristic values for varying goal in $G$. Then, $\text{Pdim}(\mathcal{U}) = \mathcal{O}\left(LW \log(U + \ell) + W \log(\ell|B|n(L + 1))\right)$. Moreover, if $h$ is trained on $P$, it follows that $\text{Pdim}(\mathcal{U}) = \mathcal{O}\left(LW \log(U + \ell') + W \log(\ell'|B'|m(L + 1))\right)$.*

## 4 TRAINING FRAMEWORK

Our goal is to learn a heuristic $h$ that can be queried many times during A* search while satisfying three key requirements: (1) **Perfect admissibility**: the heuristic must satisfy $h(s) \leq h^*(s)$ for every state $s \in V$; (2) **High average value**: among admissible heuristics, those with higher mean values, defined as $\frac{1}{|S|} \sum_{s \in S} h(s)$, provide stronger guidance for search; and (3) **Fast inference**: the size of the model to calculate $h$, denoted as $|h|$, should be as small as possible to minimize per-call latency and memory overhead during search. These objectives can be expressed as the optimization problem

$$\max_h \; \frac{1}{|S|} \sum_{s \in S} h(s) \quad \text{s.t.} \quad h(s) \le h^*(s) \; (\forall s \in S), \qquad |h| \text{ is minimized.} \qquad (4)$$

**Framing the Task as Ordinal Classification.** Under Assumption 4 the problem can be treated as a classification task where each heuristic value $0, 1, \ldots, \ell$ forms a class. The cross-entropy (CE) loss,

$$CE \; = \; -\frac{1}{N} \sum_{i=1}^{N} \sum_{k=1}^{\ell} y_k^{(i)} \log p_k^{(i)}, \qquad p_k^{(i)} = \frac{\exp(x_k^{(i)})}{\sum_{j=1}^{\ell} \exp(x_j^{(i)})},$$

optimises accuracy but ignores the *order* among classes and treats under- and over-estimation equally. Because we prefer underestimation to overestimation, we introduce a new loss function called *Cross-Entropy Admissibility (CEA)*:

$$CEA \; = \; -\frac{1}{N} \sum_{i=1}^{N} \log\left( \sum_{k=1}^{h_i^*} \left( \frac{k}{h_i^*} \right)^{\beta} p_k^{(i)} \right) \; + \; \eta \left[ -\log p_{h_i^*}^{(i)} \right]. \qquad (5)$$

The *first term* reallocates probability mass to all classes $k \le h_i^*$; the weight $(k/h_i^*)^{\beta}$ decreases as $k$ moves farther below the true class. The parameter $\beta > 0$ balances admissibility (smaller $\beta$) against heuristic strength (larger $\beta$). The *second term* is a CE penalty, scaled by $\eta$, that sharpens the distribution around the true class. It discourages the model from assigning high probability to an inadmissible class even when most mass lies on admissible ones. Choosing $\eta$ so that $\eta \ll 1$ maintains the dominance of the first (admissibility) term while still penalizing low probability on the true class. With this loss, the unique global optimum is achieved when $p_{h_i^*}^{(i)} = 1$ for every sample, fulfilling both admissibility and maximal average heuristic.

**Delta heuristic.** PDBs can have imbalanced distributions of states and heuristic values. In the 6-edge Rubik's-Cube PDB, more than $86\%$ of the states fall in classes 7 and 8. Because these classes have large heuristic values, a model that over-predicts them is likely to violate admissibility on states with lower heuristics. This can be improved using a *delta heuristic* $h_\Delta$ (Sturtevant et al., 2017). Instead of storing distances in a single PDB, we store a small *base* PDB with a pattern that is a subset of the full PDB and store only the *difference* between these PDBs $\Delta = h_{\text{large}} - h_{\text{base}}$. At inference time, the final heuristic is reconstructed as $h_{\text{large}}(s) = h_{\text{base}}(s) + \Delta(s)$. Table 3 (Appendix G) shows that subtracting a 4-edge PDB from the 6-edge PDB significantly shifts the class imbalance.

## 5 EXPERIMENTS

In this section, we aim to answer the following key questions: *(1) How effective is the proposed training framework in learning strong admissible heuristics? (2) How robust is the proposed loss function to hyperparameter choices? (3) What is the trade-off between model complexity and the strength of the learned heuristic? (4) How does the bound on generalization error behave as the number of training instances increases?*

To evaluate these questions, we focus on the $3 \times 3$ Rubik's Cube, where, to our knowledge, no previous machine learning methods have learned an admissible heuristic. We selected four PDBs with distinct characteristics: 8-Corner, $\Delta(6,4)$-Edge, 6-Edge, and 7-Edge. All PDBs are sourced from the HOG2 repository[1]. The summary statistics for the PDBs are presented in Table 1, and the complete heuristic distributions are provided in Table 3 (Appendix G). For learning heuristic models, we adopt a neural network architecture based on the ResNet model (He et al., 2016). A complete description of the model architecture and the selected hyperparameters is presented in Appendix G.

---

[1]https://github.com/nathansttt/hog2/tree/PDB-refactor

**Training and Sampling Strategy.** Looking at the PDB heuristic distributions in Table 3 (Appendix G), there is a massive class imbalance in all PDBs. If we uniformly sample training batches, the model overfits the most populated classes. To avoid this, we handle imbalance at the *sampling* stage. Mini-batches are constructed by uniformly sampling within each heuristic class, and the number of sampled states from each class is proportional to its size. Our setting also differs from standard supervised learning. In a typical setup, we train on a subset and aim to generalize to a small test set that represents a much larger unseen population. Here, each PDB contains *all states* of a graph, and a ranking function maps every Rubik's Cube state uniquely to a PDB state. If a model predicts an admissible heuristic for every state in the PDB, we obtain a fully admissible heuristic for Rubik's Cube. Therefore, the goal is to compress the information in this large dataset into a small model while losing as little information as possible. To this end, we sample training batches from the full dataset for a fixed number of epochs. Many individual states are never seen during training, due to both dataset size and randomness in sampling, but training still reflects the full data distribution.

**Post-hoc weight pruning.** One of the constraints in Equation 4 is the model size $|h|$, since fast heuristic evaluation during search is essential. A common approach for improving inference efficiency is *post-hoc weight pruning* and related compression techniques (Han et al., 2015; Micikevicius et al., 2017; Krishnamoorthi, 2018). In our work, we adopt a simple but effective variant: we train all models in 32-bit precision but perform inference in 16-bit precision, substantially reducing latency during search. We also experimented with 8-bit quantized inference; however, the accuracy loss was significant, so we chose not to consider it further.

Table 1: Summary statistics for the PDBs.

| PDB | Avg. Heuristic | Number of States | PDB Size (MB) |
|---|---|---|---|
| 4-edge | 6.75 | 3,041,280 | 1.52 |
| 6-edge | 7.65 | 42,577,920 | 21.29 |
| 8-corner | 8.76 | 88,179,840 | 44.09 |
| 7-edge | 8.51 | 510,935,040 | 255.47 |

Table 2: Comparison between learned neural network heuristics and compressed PDBs.

| Heuristic Type | Pattern | Avg. Heuristic | Overestimation Rate | Model Size (MB) | Compression Rate |
|---|---|---|---|---|---|
| NN + CEA loss | | 7.45 | $2 \times 10^{-5}$ | 3.75 | 68.12× |
| NN + CE loss | 7-edge | 8.44 | $1.4 \times 10^{-2}$ | 3.75 | 68.12× |
| Compressed PDB | | 6.83 | 0 | 3.65 | 70.00× |
| NN + CEA loss | | 8.76 | $3 \times 10^{-7}$ | 1.89 | 23.32× |
| NN + CE loss | 8-corner | 8.76 | $2 \times 10^{-3}$ | 1.89 | 23.32× |
| Compressed PDB | | 6.84 | 0 | 1.91 | 23.00× |
| NN + CEA loss | | 6.92 | $9 \times 10^{-5}$ | 1.95 | 10.91× |
| NN + CE loss | 6-edge | 7.46 | $9 \times 10^{-2}$ | 1.95 | 10.91× |
| Compressed PDB | | 6.68 | 0 | 1.93 | 11.03× |
| NN + CEA loss | | 1.31 | $3 \times 10^{-6}$ | 3.20 | 6.65× |
| NN + CE loss | $\Delta(6,4)$-edge | 1.89 | $15 \times 10^{-2}$ | 3.20 | 6.65× |
| Compressed PDB | | 1.05 | 0 | 3.04 | 7.00× |

**Evaluating Learned Heuristics.** We compare the heuristic learned with CEA loss function against two baselines: (1) the learned heuristic using CE loss, and (2) a compressed PDB constructed using the min compression technique. The compression factor was chosen so that the compressed PDB and both NN models occupy the same amount of memory. For the NN models, we used the same hyperparameters and allocated the same number of training epochs for both loss functions. The CEA-specific parameters, $\beta$ and $\eta$, are tuned per model; Appendix J details a general tuning procedure applicable to any model.

The most critical property of a learned heuristic for achieving optimality is its overestimation rate, which should ideally be zero. However, assigning zero to all states, though admissible, offers no useful guidance. Our goal, therefore, is to learn a heuristic that is both admissible and highly

informative. Our results are summarized in Table 2. We achieved an overestimation rate which is nearly zero, indicating a fully admissible heuristic, for the 8-corner and $\Delta(6, 4)$-Edge PDBs, and only a few thousand overestimated states for the 6-edge and 7-edge PDBs. The CEA loss achieved a fully admissible heuristic in the 8-corner PDB while matching the average heuristic of the original PDB, demonstrating no information loss. Across all PDBs, the overestimation rate for our loss function is nearly $10^4\times$ smaller than that obtained with the CE loss. The comparison of the distributions of overestimated states for both loss functions across all PDBs is presented in Figures 6–9 in Appendix H. These findings suggest that CE loss is ill-suited for learning admissible heuristics in domains satisfying Assumption 4, where the heuristic learning problem can be formulated as a classification task. We attribute the challenges faced by the CE loss in learning admissible heuristics to two main factors: (1) the dataset's large size and (2) the sparse representation of each state, which includes only the location and rotation of each cubie. In deep learning classification tasks for large datasets (Sun et al., 2017), there are usually no constraints on model size, and the representation for each state is richer.

Compared to a compressed PDB built with the min–compression technique, CEA achieves a significantly higher average heuristic on all PDBs, even though a few thousand overestimated states remain in the 6-edge and 7-edge cases. This suggests that CEA preserves more information than classical compression methods such as min–compression. For the 8-corner PDB, when the learned model is used in search to assess its strength, the number of nodes generated was less than half of that for the size-matched compressed PDB heuristic (Futuhi & Sturtevant, 2025). To evaluate how far we can reduce model size while maintaining admissibility and accuracy, we conduct an experiment on the 8-corner PDB (Appendix I). Remarkably, we achieve a $51\times$ compression relative to the original PDB while maintaining performance comparable to the results in Table 2.

**Empirical analysis of the generalization error.** Up to this point, our experiments have focused on the final overestimation rate of the learned heuristic. Since our sample-complexity analysis establishes a relationship between the generalization error term and the number of training instances, we conduct an additional experiment to empirically examine this behavior. Specifically, we evaluate how the generalization error associated with the inadmissibility functions in Theorem 4 changes with number of training instances. Figure 2 presents the results for 8-corner PDB. We observe that the generalization error decreases as the number of training instances $N$ increases, following a sublinear decay pattern that aligns with the

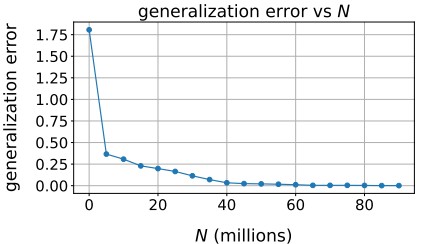

Figure 2: Generalization error vs. training size for the 8-corner PDB.

theoretical rate of $\tilde{\mathcal{O}}\left(\hat{H}\sqrt{(n + \log(1/\delta))/N}\right)$. This confirms the expected $1/\sqrt{N}$ decay predicted by our sample-complexity analysis.

## 6    CONCLUSION AND FUTURE WORK

This paper lays a foundation for learning admissible heuristics that are both effective in practice and backed by theory. We formulate heuristic learning as an optimization problem with explicit constraints on model size, heuristic strength, and admissibility. We introduce Cross-Entropy Admissibility (CEA), a loss that improves accuracy while directly penalizing any admissibility error. In experiments on $3 \times 3$ Rubik's-Cube PDBs, CEA reduced the ratio of inadmissible states to below $10^{-6}$ and matched or outperformed classical compression techniques On the theory side, we tightened the sample-complexity bounds by leveraging the exponential reduction offered by PDB abstractions and the graph structure commonly seen in heuristic-search tasks. When the hypothesis class is restricted to neural networks, the bound depends primarily on network depth and width rather than graph size. We provide the first generalization guarantees for goal-dependent heuristics. We also introduce a new bound on expected suboptimality using the maximum inadmissibility encountered at any state on the optimal path. The focus of future work will be on finding the most effective ways adapt both search and learning to work together while providing solution quality guarantees.

## ACKNOWLEDGMENTS

This work was supported by the National Science and Engineering Research Council of Canada Discovery Grant Program and the Canada CIFAR AI Chairs Program.

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

# Appendix

C­ONTENTS

## A  RELATED WORK

There has been significant effort to learn good heuristics for heuristic search algorithms (Chen et al., 2023; Numeroso et al., 2022; Greco et al., 2022; Chen et al., 2024b; Ferber et al., 2020; Yu et al., 2020; Kim & An, 2020; ús Virseda et al., 2013; Heller et al., 2022; Agostinelli et al., 2019; Kirilenko et al., 2023; Chen et al., 2024a; Hao et al., 2024; Bettker et al., 2024). One branch of work uses *Graph Neural Networks* to better learn the heuristic function by exploiting the graph structure of the problem (Pándy et al., 2022; Shen et al., 2020). In heuristic search problems, the heuristic value of a state can often be approximated from its neighbors' values, as they differ only by the cost of the connecting edge. Consequently, one can employ *bootstrapping* methods such as TD-learning (Sutton, 1988) to learn the heuristic value for each state (Ferber et al., 2022; Agostinelli et al., 2024). Veerapaneni et al. (2023) take a different approach than learning the global cost-to-go values, since these can be time-consuming to train and difficult to generalize to new problems. Instead, Veerapaneni et al. (2023) train *local heuristics* that estimate the cost to escape from small regions for the robot, and later combine these local heuristics with a global heuristic to reduce node expansions. Because data collection for the local heuristic can be slow for large graphs, Veerapaneni et al. (2024) propose an efficient data collection approach that leverages the combinatorial nature of tasks. Another branch of work considers directly learning the policy that decides which node to expand next rather than learning heuristic values (Orseau & Lelis, 2021; Gomoluch et al., 2020; Feng et al., 2022; Choudhury et al., 2018). For instance, Choudhury et al. (2018) train a policy using *imitation learning* that bases its decisions solely on the portion of the search space uncovered so far.

Garrett et al. (2016) propose focusing on the ordering of states induced by the heuristic, rather than learning exact heuristic values. They employ Rank Support Vector Machines (RankSVM) by using the number of incorrectly ordered pairs of states in each problem as the loss function. Later on, Chrestien et al. (2024) provide theoretical proof that training heuristic functions to produce correct rankings is sufficient for optimal performance, and explain why learning the exact cost-to-go values

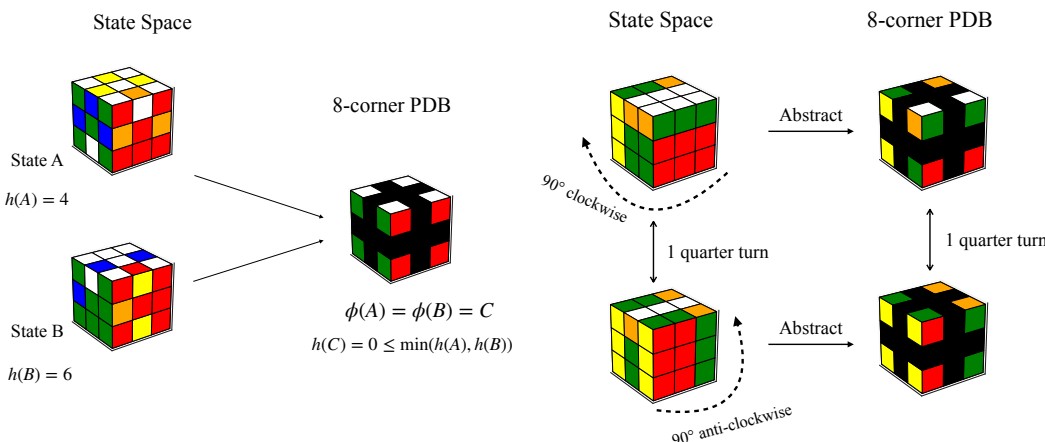

Figure 3: Comparison between the full state-space graph of the $3 \times 3$ Rubik's Cube and the abstraction generated using only the eight corner cubies.

---

**Algorithm 1** A* algorithm

---

1: OPEN $= \{s_{init}\}$, CLOSED $= \emptyset$, Parent$(s_{init}) = $ null and $g(s_{init}) = 0$.
2: **while** OPEN is not empty **:**
3:     $s \leftarrow \text{argmin}\{g(v) + h(v)|v \in \text{OPEN}\}$             ▷ selection step.
4:     **if** $s = goal$ **:**
5:         **return** solution path using Parent$(s)$ recursively.
6:     **for** each child $s'$ of $s$ **:**
7:         $g_{\text{new}} \leftarrow g(s) + c(s, s')$.
8:         **if** $s' \notin \text{OPEN} \cup \text{CLOSED}$ **:**
9:           $g(s') \leftarrow g_{\text{new}}$, and OPEN $\leftarrow$ OPEN $\cup \{s'\}$.
10:        **else if** $s' \in \text{OPEN}$ and $g_{\text{new}} < g(s')$ **:**
11:          $g(s') \leftarrow g_{\text{new}}$ and Parent$(s') \leftarrow s$.
12:        **else if** $s' \in \text{CLOSED}$ and $g_{\text{new}} < g(s')$ **:**
13:          $g(s') \leftarrow g_{\text{new}}$ and Parent$(s') \leftarrow s$.
14:          Move $s'$ from CLOSED to OPEN.        ▷ node reopening.
15:     Move $s$ from OPEN to CLOSED.

---

via mean-squared error regression can be unnecessarily difficult; their experiments on a wide range of tasks also show that ranking-based loss functions outperform regression-based ones. Subsequently, Chrestien et al. (2022) propose the $L^*$ loss, which upper-bounds the number of expanded nodes by ensuring that states on the optimal path have lower heuristic values than those off it. Li et al. (2022) and Agostinelli et al. (2021) investigate the admissibility of learned heuristics, guaranteeing full and approximate admissibility, respectively. Núñez-Molina et al. (2024) model the learned heuristic as a Truncated Gaussian, with an admissible heuristic serving as the lower bound. Balcan et al. (2021a) introduce a theoretical framework that provides generalization guarantees for data-driven algorithms by bounding the performance gap between training data and unseen data for algorithms relying on learned parameters. This framework is further applied to establish sample-complexity bounds for general tree-search algorithms (Balcan et al., 2021b; 2022) and for GBFS/A* (Sakaue & Oki, 2022). Cheng et al. address the sample complexity of branch-and-cut problems by modeling the parameter space with neural networks. They also derive a sample complexity bound for learning instance-dependent parameters.

## B   PSEUDOCODE FOR THE A* ALGORITHM

In this section, we provide an overview of the general procedure of $A^*_h$, as shown in Algorithm 1.

## C   COMPARING THE PDB ABSTRACTION TO THE FULL GRAPH

Figure 3 compares the abstracted state space (8-corner PDB) with the main graph and highlights two basic properties. First, all states in the main graph that map to the same abstract state receive the same heuristic value—namely, the shortest-path distance in the abstracted space to the abstract goal—which is a lower bound on their true distances; thus admissibility is guaranteed. Second, if two states are connected by a move in the main graph, then their images in the abstracted space are either the same state or adjacent states by the *same* move, so connectivity is preserved under the abstraction.

## D   PROOFS FOR HEURISTIC FUNCTIONS IN $\mathbb{R}^n$

In this section, we provide proofs for theoretical results that assume $h \in \mathbb{R}^n$.

*Proof of Lemma 1.* Suppose we have two heuristic functions $h$ and $h'$ that satisfy the condition of Lemma 1. Consider two states $v_i$ and $v_j$ such that $A_h^*$ selects $v_i$ for expansion before $v_j$. By definition of the selection rule, this implies one of the following:

1.  $g(v_i) + h(v_i) < g(v_j) + h(v_j)$,

2.  $g(v_i) + h(v_i) = g(v_j) + h(v_j)$ and, under the tie-breaking rule specified by Assumption 2, $A_h^*$ chooses $v_i$ over $v_j$.

In either case, we derive: $g(v_i) + h(v_i) - h(v_j) \leq g(v_j)$. By Lemma 1, we have $h(v_i) - h(v_j) = h'(v_i) - h'(v_j)$. Substituting this into the above inequality gives

$$g(v_i) + h'(v_i) - h'(v_j) \leq g(v_j) \implies g(v_i) + h'(v_i) \leq g(v_j) + h'(v_j).$$

Hence, under $A_{h'}^*$, the same selection decision is made, choosing $v_i$ instead of $v_j$. Since this reasoning applies to all selection steps, $A_h^*$ and $A_{h'}^*$ perform identical, implying $u_h(x) = u_{h'}(x)$ for every $x \in \Pi$. $\qquad\square$

*Proof of Theorem 1.* From Lemma 1, any two heuristic functions that induce the same pairwise value differences for every pair of vertices in $V$ result in identical performance. Our aim is thus to partition $\mathbb{R}^n$ into regions in which the relative differences $h(v_i) - h(v_j)$ remain consistent for all vertex pairs $(v_i, v_j)$, guaranteeing the same performance for any $x \in \Pi$. Recall that vertices in the open list OPEN are ranked by their $f$ values, where $f(v) = g(v) + h(v)$. Under Assumption 4 with $c_0 = 1$, consider a scenario in which $g(v_1) = 2$ and $g(v_2) = 4$. Then any heuristic satisfying $h(v_1) - h(v_2) \geq 2$ will favor $v_1$ over $v_2$ in OPEN. However, suppose a different instance or an update in the search reduces $g(v_1)$ from 2 to 1. In this case, $h(v_1) - h(v_2) = 3.5$ favors $v_1$, whereas $h(v_1) - h(v_2) = 2.5$ favors $v_2$, even though both heuristic differences exceed 2. To capture all such distinctions, we consider $2n$ hyperplanes for each ordered pair $(v_i, v_j)$:

$$h(v_i) - h(v_j) = \pm 1c_0, \; \pm 2c_0, \; \ldots, \; \pm nc_0.$$

Between any two of these hyperplanes, the ordering between $v_i$ and $v_j$—considering all feasible $g(\cdot)$ values—remains unchanged. Above a boundary hyperplane such as $h(v_i) - h(v_j) = nc_0$, no further hyperplanes are necessary, since $g$ values cannot exceed $nc_0$. Consequently, we have in total $2n \cdot \binom{n}{2}$ hyperplanes. By the *Shi arrangement* (Orlik & Terao, 2013), these hyperplanes partition $\mathbb{R}^n$ into $\mathcal{O}\big((2n+1)^n\big)$ regions. Each region corresponds to a unique assignment of heuristic differences for all vertex pairs, which in turn induces a unique ranking of vertices in OPEN, thus a unique performance outcome on any instance $x \in \Pi$. To shatter $N$ instances, we require at least $2^N$ such unique $(u_{x_1}, u_{x_2}, ..., u_{x_N})$ tuples; hence, $\mathcal{O}\big((2n+1)^n\big) \geq 2^N$. Solving for the largest $N$ in terms of $n$ yields the upper bound $\text{Pdim}(\mathcal{U}) = \mathcal{O}(n \log n)$. $\qquad\square$

*Proof of Theorem 2.* The dataset $P$ partitions the set $V$ of $n$ vertices into $m$ groups $\mathcal{M}_1, \mathcal{M}_2, \ldots, \mathcal{M}_m$, each containing vertices that correspond to the same abstract state. Consequently, the trained heuristic satisfies $h(v_i) = h(v_j) \;\; \forall \, v_i, v_j \in \mathcal{M}$ for any $\mathcal{M} \in P$. To compute $\text{Pdim}(\mathcal{U})$, one might naively substitute $m$ for $n$ in Theorem 1; however, the $g$ values still come from $G$, so there remain $2n$ hyperplanes for any pair of vertices. Total number of hyperplanes is $2n \cdot \binom{m}{2}$

because there are $m$ groups, and all vertices within the same group have the same $g$ cost. These hyperplanes partition the space $\mathbb{R}^m$ into $\mathcal{O}\big((2n+1)^m\big)$ regions. To shatter $N$ instances, we require at least $2^N$ such regions. Setting $\mathcal{O}\big((2n+1)^m\big) \geq 2^N$ and solving for $N$ in terms of $n$ yields $\mathrm{Pdim}(\mathcal{U}) = \mathcal{O}\big(m \log n\big)$.

To compare the bound achieved by a PDB dataset with the one in Theorem 1, we need to examine the effect of replacing the main graph $G$ with the PDB graph $P$. In the worst-case scenario, where the PDB does not abstract away any features (i.e., $|G| = |P|$ or $m = n$), there is no improvement over the $\mathcal{O}(n \log n)$ bound. However, PDBs often yield an exponential reduction in the size of the graph, so that $m = \mathcal{O}(\sqrt[c]{n})$. In this case, the resulting bound improves to $\mathcal{O}(\sqrt[c]{n} \log n)$. For instance, in an exponential state space with branching factor $b$ and depth $d$, where $n = b^d$, the root index $c$ is bounded by $\mathcal{O}(d)$. $\qquad\square$

## E  EXPECTED SUBOPTIMALITY OF A* WITH REOPENINGS

In this section, we provide the proof of Theorem 3, restated below. Before beginning the proof, we introduce the necessary terminology. We have already defined $\mathrm{INC}_h(p, c)$ for an edge from $p$ to $c$ as the inconsistency of the heuristic $h$ for this edge: $\mathrm{INC}_h(p, c) = \max\big\{ h(p) - h(c) - c(p, c), \, 0 \big\}$. For simplicity, assume that for every $h \in \mathbb{R}^n$, we have $h(goal) = 0$. Let $g^*(n)$ denote the cost of an optimal path from $n_{\mathrm{start}}$ to $n$. With this notation, $g^*(n, m)$ is the cost of an optimal path from $n$ to $m$. Throughout the search, the $g$-costs maintained by the algorithm are upper bounds on the true optimal costs. Hence, if $g_t(n)$ is the $g$-cost for vertex $n$ after $t$ iterations, we have $g_t(n) \geq g^*(n)$. We define the difference between $g_t(n)$ and $g^*(n)$ as the *g-cost error*, denoted $g_t^\delta(n) = g_t(n) - g^*(n)$. Because we only update $g_t(n)$ when we discover a strictly better path to $n$, both $g_t(n)$ and $g_t^\delta(n)$ are non-increasing for every $n \in V$.

**Theorem.** *Let $x \in \Pi$ be a problem instance, and let $P_{\mathrm{opt}} = v_0, v_1, \ldots, v_k$ be its optimal solution path with cost $C^*(x)$. Suppose A\* is allowed to reopen nodes. Then the cost $C_h(x)$ of any solution returned by A\* satisfies*

$$C_h(x) \; - \; C^*(x) \; \leq \; \max_{v \in P_{\mathrm{opt}}} \Big[ h(v) \; - \; h^*(v) \Big].$$

*Proof.* To prove the above theorem, we first state the following lemma from Hart et al. (1968). We omit its proof, which is straightforward and appears in the original work.

**Lemma 2.** *Let $P_{\mathrm{opt}}$ be an optimal solution path to a given problem. At any time prior to the expansion of a goal node by A\*, there is a node from $P_{\mathrm{opt}}$ which is in* OPEN.

Suppose that for a problem instance $x \in \Pi$, the optimal path is $P_{\mathrm{opt}} = v_0, v_1, \ldots, v_k$, and that the goal node $v_k$ is selected for expansion at iteration $t$. Let $v_i$ be the node from $P_{\mathrm{opt}}$ in OPEN at this time. Since we select $v_k$ instead of $v_i$, we have:

$$f(v_i) > f(v_k), \tag{1}$$
$$g_t(v_i) + h(v_i) > g_t(v_k) + h(v_k), \tag{2}$$
$$g^*(v_i) + g_t^\delta(v_i) + h(v_i) > g_t(v_k), \tag{3}$$
$$g^*(v_i) + g_t^\delta(v_i) + h(v_i) + h^*(v_i) - h^*(v_i) > C, \tag{4}$$
$$C^* + g_t^\delta(v_i) + h(v_i) - h^*(v_i) > C, \tag{5}$$
$$C - C^* < g_t^\delta(v_i) + \big[ h(v_i) - h^*(v_i) \big]. \tag{6}$$

Here, (2) follows from the definition of $f(\cdot)$, (3) follows from the definition of $g_t(\cdot)$, and (5) uses the fact that the optimal path can be split at any node $n \in P_{\mathrm{opt}}$ into the optimal path from $n_{\mathrm{start}}$ to $n$ and from $n$ to $v_k$. Inequality (6) thus indicates that the suboptimality can be bounded by the sum of the $g$-cost error for $n_i$ and the difference $h(n_i) - h^*(n_i)$.

Valenzano et al. (2014) showed that for A* without reopening, we have

$$g_t^\delta(n_i) \le \sum_{j=1}^{i-1} \text{INC}_h(v_j, v_{j+1}),$$

$$h(n_i) - h^*(n_i) \le \sum_{j=i}^{k-1} \text{INC}_h(v_j, v_{j+1}),$$

$$g_t^\delta(n_i) + h(n_i) - h^*(n_i) \le \sum_{j=1}^{k-1} \text{INC}_h(v_j, v_{j+1}).$$

To complete the proof of Theorem 3, we must show that if A* allows reopening nodes, then $g_t^\delta(n_i) = 0$. In that case:

$$C - C^* < g_t^\delta(n_i) + [\,h(n_i) - h^*(n_i)\,], \tag{7}$$
$$C - C^* < [\,h(n_i) - h^*(n_i)\,], \tag{8}$$
$$C - C^* < \max_{v \in P_{\text{opt}}} [\,h(v) - h^*(v)\,]. \tag{9}$$

Inequality (9) follows from the definition of the max function. The next theorem implies this condition and thus completes the proof of Theorem 3.

**Theorem 8.** *Let $P_{opt} = v_0, v_1, \ldots, v_k$ be an optimal solution path to a given problem. If at iteration $t$ the goal node $v_k$ is selected for expansion, there will be a node $v_i$ from $P_{opt}$ which is in OPEN such that $g_t^\delta(v_i) = 0$.*

The proof is by induction on the number of iterations (i.e., node expansions), denoted by $t$. If $t = 0$ and the goal is selected for expansion, it means $P_{\text{opt}} = v_0$ and the statement is vacuously true. If $t = 1$, we have OPEN $= \{$children of $v_0\}$ and CLOSED $= \{v_0\}$. Since the goal is selected for expansion, it must be one of $v_0$'s children. Now, if $P_{\text{opt}} = v_0, v_1$, the statement is correct since $v_1$ from the optimal path is in OPEN and $g_t^\delta(v_1) = 0$. Suppose the goal does not appear immediately after $v_0$ in $P_{\text{opt}}$. In this case, $v_1$ is not the goal, but it connects $v_0$ to the rest of $P_{\text{opt}}$. Since the edge $(v_0, v_1)$ is on $P_{\text{opt}}$, we can conclude that $g^*(v_1) = g^*(v_0, v_1) = c(v_0, v_1)$, and thus the statement is correct.

We assume that for all iterations from 1 to $n$, if the goal is selected for expansion, there is a node $v_i$ from $P_{\text{opt}}$ in OPEN such that $g_t^\delta(v_i) = 0$. Now consider iteration $n + 1$ where the goal $v_k$ is selected for expansion. This shows that the node expanded at iteration $n$ was not the goal. If the expanded node at iteration $n$ is not $v_i$, then the statement remains correct for iteration $t + 1$ as well, because $v_i$ from the previous iteration is still in OPEN. If the expanded node is $v_i$, it is moved to CLOSED, and there are three cases for its child $v_{i+1}$ on $P_{\text{opt}}$:

- $v_{i+1}$ is already on OPEN.

- $v_{i+1}$ is not explored. In this case, we add $v_{i+1}$ to OPEN.

- $v_{i+1}$ is already on CLOSED. In this case, we move $v_{i+1}$ back to OPEN.

In all three cases, we update the $g$-cost of $v_{i+1}$ using $v_i$'s $g$-cost. This is because every subpath on $P_{\text{opt}}$ between any two nodes $p, c$ has the optimal $g$-cost $g^*(p, c)$. As a result, $v_{i+1}$ will have the optimal $g$-cost. Having handled all scenarios for iteration $t + 1$, the statement is correct by induction.

$\square$

*Proof of Theorem 4.* To apply Proposition 1 to $\hat{\mathcal{U}}$, we require $\Psi_h(\cdot) : \Pi \to [0, \hat{H}]$. This is not always possible if the heuristic function can assign $h(v) = \infty$ to some vertex $v$. To avoid this, we assume that $\forall v \in V, h(v) \ne \infty$ if $h^*(v) \ne \infty$, which means the learned heuristic cannot treat

any node from which the goal is reachable as a dead-end. We know that $\hat{H}$ is not greater than the upper bound derived on expected suboptimality without reopening. In Valenzano et al. (2014), a worst-case graph and Martelli's subgraphs (Martelli, 1977) show that without reopenings, the bounds on expected suboptimality are tight. Hence, $\hat{H}$ is not too large relative to suboptimality. For this theorem, measuring the pseudo-dimension of $\mathcal{H}$ directly is challenging, while it is simpler to measure it for the *dual class* of $\mathcal{H}$. We introduce Assouad's dual class of $\mathcal{H}$ as follows.

**Definition 4** (Assouad (1983))**.** *Given a class, $\mathcal{H} \subseteq \mathbb{R}^{\mathcal{Y}}$, of functions $h : \mathcal{Y} \to \mathbb{R}$, the dual class of $\mathcal{H}$ is defined as $\mathcal{H}^* = \left\{ h_y^* : \mathcal{H} \to \mathbb{R} \mid y \in \mathcal{Y} \right\}$ such that $h_y^*(h) = h(y)$ for each $y \in \mathcal{Y}$.*

We now introduce a class known as $(\mathcal{F}, \mathcal{B}, K)$-*piecewise decomposable*, which has a piecewise structure as formalized in Definition 5. If the dual class of $\mathcal{H}$ is $(\mathcal{F}, \mathcal{B}, K)$-piecewise decomposable, we can use Proposition 2 to bound the pseudo-dimension of $\mathcal{H}$.

**Definition 5** (Balcan et al. (2021a, Definition 3.2))**.** *A class, $\mathcal{H} \subseteq \mathbb{R}^{\mathcal{Y}}$, of functions is $(\mathcal{F}, \mathcal{B}, K)$-piecewise decomposable for a class $\mathcal{B} \subseteq \{0,1\}^{\mathcal{Y}}$ of boundary functions and a class $\mathcal{F} \subseteq \mathbb{R}^{\mathcal{Y}}$ of piece functions if the following condition holds: for every $h \in \mathcal{H}$, there exist $K$ boundary functions $b^{(1)}, \ldots, b^{(K)} \in \mathcal{B}$ and a piece function $f_{\boldsymbol{b}}$ for each binary vector $\boldsymbol{b} \in \{0,1\}^K$ such that for all $y \in \mathcal{Y}$, it holds that $h(y) = f_{\boldsymbol{b}_y}(y)$ where $\boldsymbol{b}_y = (b^{(1)}(y), \ldots, b^{(K)}(y)) \in \{0,1\}^K$.*

**Proposition 2** (Balcan et al. (2021a, Theorem 3.3))**.** *Let $\mathcal{U} \subseteq \mathbb{R}^{\Pi}$ be a class of functions. If $\mathcal{U}^* \subseteq \mathbb{R}^{\mathcal{U}}$ is $(\mathcal{F}, \mathcal{B}, K)$-piecewise decomposable with a class $\mathcal{B} \subseteq \{0,1\}^{\mathcal{U}}$ of boundary functions and a class $\mathcal{F} \subseteq \mathbb{R}^{\mathcal{U}}$ of piece functions, the pseudo-dimension of $\mathcal{U}$ is bounded as follows:*

$$\mathrm{Pdim}(\mathcal{U}) = \mathrm{O}((\mathrm{Pdim}(\mathcal{F}^*) + \mathrm{VCdim}(\mathcal{B}^*)) \log(\mathrm{Pdim}(\mathcal{F}^*) + \mathrm{VCdim}(\mathcal{B}^*)) + \mathrm{VCdim}(\mathcal{B}^*) \log K).$$

Now, we begin our proof by defining the function classes $\mathcal{B}$ and $\mathcal{F}$. We consider that $\mathcal{B} = \left\{ b_{h,z_0} : \hat{\mathcal{U}} \to \{0,1\} \mid h \in \mathbb{R}^n, z_0 \in \mathbb{R} \right\} \subseteq \{0,1\}^{\hat{\mathcal{U}}}$ and $\mathcal{F} = \left\{ f_h : \hat{\mathcal{U}} \to \mathbb{R} \mid h \in \mathbb{R}^n \right\} \subseteq \mathbb{R}^{\hat{\mathcal{U}}}$ to be classes of boundary and piece functions, respectively. We first prove that $\hat{\mathcal{U}}^*$ is $(\mathcal{F}, \mathcal{B}, \mathrm{O}(n))$-piecewise decomposable. Fix any $\Psi_x^* \in \hat{\mathcal{U}}^*$. This choice determines a unique instance $x \in \Pi$ and its optimal solution $P_{\mathrm{opt}}(x) \subseteq E$. Let $K = |V| = \mathrm{O}(n)$. We define $K$ boundary functions of the form $b^{(v)}(h) = \mathbb{I}(h_v - h_v^* > 0)$ for each vertex $v \in V$. These boundary functions partition $\mathbb{R}^n$ into regions such that, in each region, $\Psi_x^*(h)$ is expressible as a function in $h$, which belongs to $\mathcal{F}$. Specifically, for a given binary vector $\boldsymbol{b}_h = \left(b^{(v)}(h)\right)_{v \in V} \in \{0,1\}^K$, define $P_h(x) \subseteq P_{\mathrm{opt}}(x)$ by $P_h(x) = \left\{ v \in P_{\mathrm{opt}}(x) : b^{(v)}(h) = 1 \right\}$. Hence $v \in P_h(x)$ whenever $h_v - h_v^* > 0$. From the definition of $\Psi_h(x)$, we obtain $\Psi_x^*(h) = \Psi_h(x) = \max_{v \in P_h(x)}\left(h_v - h_v^*\right)$. This quantity is piecewise linear in $h$, so we can pick $f_{\boldsymbol{b}_h} \in \mathcal{F}$ such that $\Psi_x^*(h) = f_{\boldsymbol{b}_h}(h)$ in that region. Since this holds for every $\boldsymbol{b}_h$ in $\{0,1\}^K$, we conclude $\Psi_x^*(h) = f_{\boldsymbol{b}_h}(h)$ for all $h \in \mathbb{R}^n$. Therefore, $\hat{\mathcal{U}}^*$ is $(\mathcal{F}, \mathcal{B}, \mathrm{O}(n))$-piecewise decomposable. Since $\mathcal{B}$ is the set of single-coordinate threshold functions in $\mathbb{R}^n$, we have $\mathrm{VCdim}(\mathcal{B}^*) = \mathrm{VCdim}(\mathcal{B}) = n$. For $\mathcal{F}$, which is a family of $\max$ functions on $\mathbb{R}^n$, each function can shatter at most $n$ instances. By (Balcan et al., 2021a, Lemma 3.10), we know $\mathrm{Pdim}(\mathcal{F}^*) \leq \mathrm{Pdim}(\mathcal{F}) = n$. Consequently, from Proposition 2, it follows that $\mathrm{Pdim}(\hat{\mathcal{U}}) = \mathrm{O}(n \log n)$.

Although we introduced a framework with fewer boundary functions, the bound here is not sharper than that of (Sakaue & Oki, 2022). The key limitation is that, even if $\mathcal{F}$ contained only constant functions, the dimension would still be constrained by $\mathrm{VCdim}(\mathcal{B})$. □

# F  PROOF OF THEOREMS USING NEURAL NETWORKS

In this section, we provide proofs for the theorems that use neural networks to represent the heuristic function.

*Proof of Theorem 5.* In many planning domains such as the Sliding Tile Puzzle (STP), TopSpin, and Rubik's Cube—the primary applications considered in this work—all the assumptions from Theorem 5 hold. By these assumptions, there are $\left\lceil \frac{D}{c_0} \right\rceil$ possible heuristic values. This allows us to

define $A_h$ with the heuristic $h = N^{\text{ReLU, softmax}} : \mathbb{R}^{w_0} \times \mathbb{R}^W \to \mathbb{R}^\ell$, where $\ell = \left\lceil \frac{D}{c_0} \right\rceil$. Since each vertex can only take one of $\ell$ heuristic values, the total number of possible heuristic functions is $\ell^n$. Thus, there are at most $\ell^n$ distinct performance measures. Setting $\mathcal{O}(\ell^n) \geq 2^N$ and solving for $N$ in terms of $n$ gives $\text{Pdim}(\mathcal{U}) = \mathcal{O}(n)$. Moreover, if the neural network is trained on the PDB dataset $P$, which reduces the induced graph size to $m$, a similar argument shows that $\text{Pdim}(\mathcal{U}) = \mathcal{O}(m)$. Note that both bounds are derived under the assumption that the model is sufficiently expressive to realize all possible $\ell^n$ or $\ell^m$ heuristic combinations, respectively. $\qquad\square$

For the next proofs, we first introduce the necessary auxiliary lemmas and definitions.

**Lemma 3** (Lemma A.1 in (Cheng et al.)). *For any $x_1, \ldots, x_n, \lambda_1, \ldots, \lambda_n > 0$, the following inequalities hold:*

$$\log x_1 \leq \frac{x_1}{\lambda_1} + \log\left(\frac{\lambda_1}{e}\right), \tag{10}$$

**Lemma 4** (Theorem A.2 in (Cheng et al.)). *Let $\mathcal{P} \subseteq \mathbb{R}^\ell$ and let $f_1, \ldots, f_t : \mathbb{R}^\ell \to \mathbb{R}$ with $t \geq \ell$ be functions that are polynomials of degree $m$ when restricted to $\mathcal{P}$. Then*

$$|\{(\text{sgn}(f_1(\mathbf{p})), \ldots, \text{sgn}(f_t(\mathbf{p}))) : \mathbf{p} \in \mathcal{P}\}| = 1, \qquad\qquad m = 0,$$

$$|\{(\text{sgn}(f_1(\mathbf{p})), \ldots, \text{sgn}(f_t(\mathbf{p}))) : \mathbf{p} \in \mathcal{P}\}| \leq \left(\frac{et}{\ell+1}\right)^{\ell+1}, \qquad\qquad m = 1,$$

$$|\{(\text{sgn}(f_1(\mathbf{p})), \ldots, \text{sgn}(f_t(\mathbf{p}))) : \mathbf{p} \in \mathcal{P}\}| \leq 2\left(\frac{2etm}{\ell}\right)^\ell, \qquad\qquad m \geq 2.$$

**Lemma 5** (Lemma A.3 in (Cheng et al.)). *Let $h : \mathcal{I} \times \mathcal{P} \to \mathbb{R}$ define a parameterized function class with $\mathcal{P} \subseteq \mathbb{R}^\ell$, and let $\mathcal{H}$ be the corresponding hypothesis class. Let $m \in \mathbb{N}$ and $R : \mathbb{N} \to \mathbb{N}$ be a function with the following property: for any $t \in \mathbb{N}$ and $I_1, \ldots, I_t \in \mathcal{I}$, there exist $R(t)$ subsets $\mathcal{P}_1, \ldots, \mathcal{P}_{R(t)}$ of $\mathcal{P}$ such that $\mathcal{P} = \cup_{i=1}^{R(t)} \mathcal{P}_i$ and, for all $i \in [R(t)]$ and $j \in [t]$, $h(I_j, \mathbf{p})$ restricted to $\mathcal{P}_i$ is a polynomial function of degree at most $m$ depending on at most $\ell' \leq \ell$ of the coordinates. In other words, the map*

$$\mathbf{p} \mapsto (h(I_1, \mathbf{p}), \ldots, h(I_t, \mathbf{p}))$$

*is a piecewise polynomial map from $\mathcal{P}$ to $\mathbb{R}^t$ with at most $R(t)$ pieces. Then,*

$$\text{Pdim}(\mathcal{H}) \leq \sup\left\{t \geq 1 : 2^{t-1} \leq R(t)\left(\frac{2et(m+1)}{\ell'}\right)^{\ell'}\right\}$$

**Lemma 6** (Lemma A.4 in (Cheng et al.)). *Let $N^{\text{ReLU}} : \mathbb{R}^d \times \mathbb{R}^W \to \mathbb{R}^\ell$ be a neural network function with $\text{ReLU}$ activation and architecture $\mathbf{w} = [d, w_1, \ldots, w_L, \ell]$ (Definition 2). Then for every natural number $t > LW$, and any $\mathbf{x}^1, \ldots, \mathbf{x}^t \in \mathbb{R}^d$, there exists subsets $\mathcal{W}_1, \ldots, \mathcal{W}_Q$ of $\mathbb{R}^W$ with $Q \leq 2^L \left(\frac{2et \sum_{i=1}^L (iw_i)}{LW}\right)^{LW}$ whose union is all of $\mathbb{R}^W$, such that $N(\mathbf{x}^j, \mathbf{w})$ restricted to $\mathbf{w} \in \mathcal{W}_i$ is a polynomial function of degree at most $L + 1$ for all $(i, j) \in [Q] \times [t]$.*

*Proof.* The proof of Lemma 6 is provided in Section 2 of (Bartlett et al., 1998), and the proof of Lemma 5 is presented in Section A of (Cheng et al.). $\qquad\square$

*Proof of Theorem 6.* Our goal is to apply Lemma 5 to heuristic functions parameterized by $\mathcal{P} = \mathbb{R}^W$. Specifically, we must show that for any $t \in \mathbb{N}$ and instances $I_1, \ldots, I_t \in \mathcal{I}$, there exist $R(t)$ subsets $\mathcal{P}_1, \ldots, \mathcal{P}_{R(t)}$ of $\mathcal{P}$ such that $\mathcal{P} = \cup_{i=1}^{R(t)} \mathcal{P}_i$ and, for all $i \in [R(t)]$ and $j \in [t]$, $u(I_j, \mathbf{p})$ restricted to $\mathcal{P}_i$ is a polynomial function of degree at most $m$ depending on at most $\ell' \leq \ell$ of the coordinates.

**Partitioning $\mathbb{R}^W$: Hidden Layers and Final Layer.** We first split $\mathbb{R}^W$ into two parts: $\mathbf{W}'$ (the parameters for the hidden layers) and $\mathbb{R}^{\ell \times w_L}$ (the parameters for the final layer and its activation). There is a one-to-one correspondence between $\mathbb{R}^W$ and $\mathbf{W}' \times \mathbb{R}^{\ell \times w_L}$, where $W' = W - \ell w_L$. By Lemma 6, the number of regions for $\mathbf{W}'$ is $\left(\frac{e\,t\,U}{W'}\right)^{W'}$, so within each such region, $N^{\text{ReLU}}\big(\text{Rep}(I_j), \mathbf{w}\big)$ is a polynomial function of degree at most $L + 1$ for all $j \in [t]$.

**Analyzing the Final Layer.** Next, we analyze $\text{Softmax}(A^{L+1}\mathbf{z}^j)$, the output of the final layer, where $\mathbf{z}^j$ denotes the hidden-layer output for the instance $I_j$ in a given region of $\mathbf{W}'$. Let $A^{L+1} \in \mathbb{R}^{\ell \times w_L}$ be the weight matrix of the final layer. Then

$$\big(\text{Softmax}(A^{L+1}\mathbf{z}^j)\big)_k \;=\; \frac{\exp\Big(\sum_{i=1}^{w_L} A_{ki}^{L+1}\,\mathbf{z}_i^j\Big)}{\sum_{k'=1}^{\ell} \exp\Big(\sum_{i=1}^{w_L} A_{k'i}^{L+1}\,\mathbf{z}_i^j\Big)}, \quad k = 1, \ldots, \ell.$$

Define

$$\theta_{ki} \;=\; \exp\big(A_{ki}^{L+1}\big), \quad \text{so that} \quad \exp\big(A_{ki}^{L+1}\,\mathbf{z}_i^j\big) \;=\; \big(\theta_{ki}\big)^{\mathbf{z}_i^j}.$$

Thus, we can rewrite:

$$\big(\text{Softmax}(A^{L+1}\mathbf{z}^j)\big)_k \;=\; \frac{\prod_{i=1}^{w_L}\big(\theta_{ki}\big)^{\mathbf{z}_i^j}}{\sum_{k'=1}^{\ell} \prod_{i=1}^{w_L}\big(\theta_{k'i}\big)^{\mathbf{z}_i^j}}.$$

**Arg Max Operation for Heuristic Prediction.** The heuristic is chosen by:

$$h(I_j) \;=\; \arg \max_{1 \le k \le \ell} \big(\text{Softmax}(A^{L+1}\mathbf{z}^j)\big)_k.$$

Because the exponential function is strictly increasing, softmax preserves the ordering of its inputs. Hence, the softmax function itself does not alter the partitioning of $\mathbb{R}^{\ell \times w_L}$ needed for counting regions. We consider $\arg\max(\text{Softmax}(\cdot))$ effectively as $\arg\max$ on $A^{L+1}\mathbf{z}^j$. Following Theorem 1, we use $\Gamma = 2\ell \cdot \binom{|B|}{2}$ hyperplanes

$$h(v_i) - h(v_j) \;=\; \pm 1\,c_0,\ \pm 2\,c_0,\ \ldots,\ \pm\ell\,c_0, \quad \forall\,(v_i, v_j) \in B,$$

to partition $\mathbb{R}^\ell$. These hyperplanes keep the coordinate order fixed within each region, implying a constant A* performance in each region. In other words, if we define polynomial functions $\psi_1^j, \ldots, \psi_\Gamma^j$ for all $A^{L+1}$ such that

$$\psi_1^j\big(\arg\max(\text{Softmax}(A^{L+1}\mathbf{z}^j))\big), \ldots, \psi_\Gamma^j\big(\arg\max(\text{Softmax}(A^{L+1}\mathbf{z}^j))\big)$$

have the same signs, then

$$\mathcal{U}\big(I_j, (\mathbf{w}, A^{L+1})\big) \;=\; \mathcal{U}\Big(I_j, \varphi_{\mathbf{w}, A^{L+1}}^{N^{\text{ReLU}}, \text{Softmax}}(\text{Rep}(I_j))\Big)$$

remains constant. Since the softmax preserves order, each $\arg\max(\text{Softmax}(A^{L+1}\mathbf{z}^j))$ can be viewed as a polynomial of degree $w_L$ in $\theta_{ki}$. The hyperplanes $\psi_1^j, \ldots, \psi_\Gamma^j$ each have degree 1. The total number of such functions across all training instances, i.e. $\psi_1^j(\arg\max(A^{L+1}\mathbf{z}^j)), \ldots, \psi_\Gamma^j(\arg\max(A^{L+1}\mathbf{z}^j))$ for $j \in [t]$, is $t\Gamma$.

**Number of Regions.** By Lemma 4, we can partition $\mathbb{R}^{\ell \times w_L}$ into at most

$$2\Big(\tfrac{2e\,t\,\Gamma\,w_L}{\ell\,w_L}\Big)^{\ell w_L} \;\le\; 2\Big(\tfrac{2e\,t\,\Gamma}{\ell}\Big)^{\ell w_L}$$

regions, within each of which $u\big(I_j, (\mathbf{w}, A^{L+1})\big)$ is constant as a function of $(\mathbf{w}, A^{L+1})$. Combining this with the regions from the hidden layers $\mathbf{W}'$, the total number of regions in $\mathbb{R}^W$ is at most

$$R(t) \;=\; 2^L\Big(\tfrac{2e\,t\sum_{i=1}^L (i\,w_i)}{L\,W}\Big)^{LW} \cdot 2\Big(\tfrac{2e\,t\,\Gamma}{\ell}\Big)^{\ell w_L} \;\le\; 2^{L+1}\Big(\tfrac{2e\,t\,U}{W}\Big)^{LW} \cdot \Big(\tfrac{2e\,t\,\Gamma}{\ell}\Big)^{\ell w_L}.$$

Within each region, $u\big(I_j, (\mathbf{w}, A^{L+1})\big)$ is a polynomial of degree at most $L+1$. Applying Lemma 5, $\text{Pdim}(\mathcal{U})$ is bounded by the largest $t \in \mathbb{N}$ such that

$$2^{t-1} \;\le\; 2^{L+1}\Big(\tfrac{2e\,t\,U}{W}\Big)^{LW} \cdot \Big(\tfrac{2e\,t\,\Gamma}{\ell}\Big)^{\ell w_L} \cdot \Big(\tfrac{2e\,t\,(L+1)}{W}\Big)^{W}.$$

Taking logarithms on both sides:

$$t - 1 \;\le\; (L+1) + LW\log\Big(\tfrac{2e\,t\,U}{W}\Big) + \ell\,w_L\log\Big(\tfrac{2e\,t\,\Gamma}{\ell}\Big) + W\log\Big(\tfrac{2e\,t\,(L+1)}{W}\Big)$$

$$= (L+1) + (LW + \ell w_L + W) \log t + \left( LW \log\left(\frac{2e\,U}{W}\right) + \ell w_L \log\left(\frac{2e\,\Gamma}{\ell}\right) + W \log\left(\frac{2e\,(L+1)}{W}\right) \right).$$

Using inequality (10) from Lemma 3, with $x_1 = t$ and an appropriate $\lambda$, we get:

$$\log t \leq \frac{t}{\lambda} + \log\left(\frac{\lambda}{e}\right).$$

Substituting and setting $\lambda = 8\,(LW + \ell w_L + W)$, after simplification we obtain:

$$t\left(1 - \frac{1}{8\log 2}\right) \leq (L+1) + \frac{(LW + \ell w_L + W)}{\log 2} \log\left(\frac{8\,(LW + \ell w_L + W)}{e}\right)$$
$$+ LW \log_2\left(\frac{2e\,U}{W}\right) + \ell w_L \log_2\left(\frac{2e\,\Gamma}{\ell}\right) + W \log_2\left(\frac{2e\,(L+1)}{W}\right).$$

Solving for $t$, we conclude:

$$t = \mathcal{O}\Big((LW + \ell w_L + W) \log(U + \ell) + W \log\big(\Gamma(L+1)\big)\Big).$$

Hence,

$$\mathrm{Pdim}(\mathcal{U}) = \mathcal{O}\Big(LW \log(U + \ell) + W \log\big(\Gamma(L+1)\big)\Big).$$

**Training on the Main Graph.** When training on the main graph $G$, we have $\Gamma = 2\,\ell \cdot \binom{|B|}{2} = \ell\,|B|^2$. Substituting this gives:

$$\mathrm{Pdim}(\mathcal{U}) = \mathcal{O}\Big(LW \log(U + \ell) + W \log\big(\ell\,|B|\,(L+1)\big)\Big).$$

**Training on the Graph Induced by P.** If we train on the graph induced by $\mathbf{P}$, the only modified parameters are $B'$ and $\ell'$. In this abstracted graph, the heuristic values and the number of generated nodes during search are bounded by those in the original graph. Thus, let $B'$ be the number of states needed to represent each training instance, and $\ell'$ be the number of heuristic classes. A similar argument shows:

$$\mathrm{Pdim}(\mathcal{U}) = \mathcal{O}\Big(LW \log(U + \ell') + W \log\big(\ell'\,|B'|\,(L+1)\big)\Big).$$

$\square$

*Proof of Theorem 7.* When the *goal* state can vary for each instance, the number of states required per instance and the number of hyperplanes in the neural network's final layer both change accordingly. Consider a fixed instance $Rep(I_j)$ with start state $s_j$. If *goal* were fixed, we would need $|B|$ states to represent this single instance. However, in this setting, the goal state can be any of the $n$ states $v_i \in V$ for $i \in [n]$.

Because heuristic values for distinct $(s_j - g_i)$ pairs are generally independent, we must account for separate hyperplanes for each such pair. Therefore, we have these hyperplanes set for every instance:

$$\text{For } (s_j - g_1): \quad h(v_i) - h(v_j) = \pm 1\,c_0,\ \pm 2\,c_0,\ \ldots,\ \pm \ell\,c_0 \quad \forall\,(v_i, v_j) \in B_1,$$
$$\text{For } (s_j - g_2): \quad h(v_i) - h(v_j) = \pm 1\,c_0,\ \pm 2\,c_0,\ \ldots,\ \pm \ell\,c_0 \quad \forall\,(v_i, v_j) \in B_2,$$
$$\vdots$$
$$\text{For } (s_j - g_n): \quad h(v_i) - h(v_j) = \pm 1\,c_0,\ \pm 2\,c_0,\ \ldots,\ \pm \ell\,c_0 \quad \forall\,(v_i, v_j) \in B_n.$$

Hence, the total number of hyperplanes is

$$\Gamma = 2\,\ell\,n \cdot \binom{|B|}{2}.$$

The rest of the analysis follows the same approach as in the fixed-goal case. For the original graph $G$, we obtain:

$$\mathrm{Pdim}(\mathcal{U}) = \mathcal{O}\Big(L W \log(U + \ell) + W \log\big(\ell\,n\,|B|\,(L+1)\big)\Big).$$

and with same argument for the graph induced by dataset $P$:

$$\mathrm{Pdim}(\mathcal{U}) = \mathcal{O}\left(LW \log(U + \ell') + W \log\left(\ell' m |B'|(L+1)\right)\right).$$

$\square$

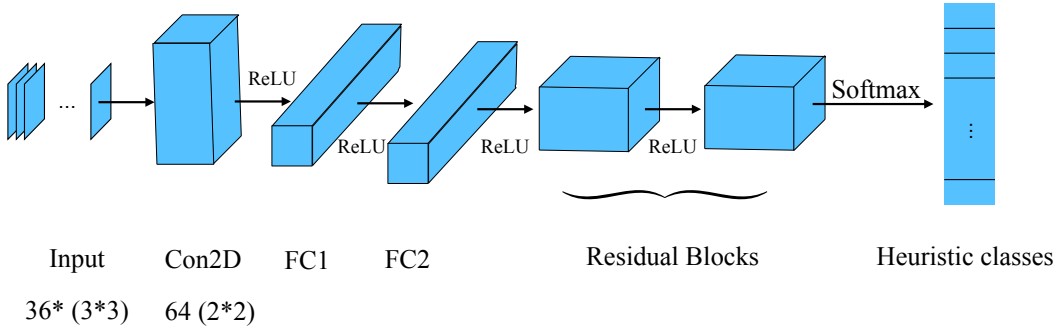

Figure 4: Neural Network structure.

## G EXPERIMENTAL SETUP

Our experiments use a ResNet architecture (He et al., 2016) (see Figure 4); the full set of training hyper-parameters appears in Table 4. Table 3 present the details for all PDBs. All runs were carried out on a server with a 32-core AMD Ryzen Threadripper 2950X CPU and two NVIDIA GeForce RTX 2080 Ti GPUs (CUDA 12.4). The NN model sizes and performances reported in Table 2 correspond to the 16-bit precision models used during search.

### G.1 STATE REPRESENTATION

For each PDB, we use one-hot encodings tailored to the type of cubies included:

- **8-Corner PDB**. Each face is encoded using six $3\times3$ channels—one per color—resulting in a total of $6 \times 6 = 36$ channels. All non-corner cubies are assumed to be in the solved state.

- **Edge PDBs**. For a PDB with $n$ edges, we construct a $3n$-channel input of size $4\times3$, consisting of: (i) *location*—a one-hot map over the 12 possible edge positions; (ii) *rotation*—a one-hot indicator set if the cubie is correctly oriented; and (iii) *goal position*—a fixed map encoding the target location for each edge.

Table 3: Heuristic distributions for the PDBs.

| $h$ | 8-corner | 7-edge | 6-edge | 6–4 (delta) |
|---|---|---|---|---|
| 0 | 1 | 1 | 1 | 1,076,354 |
| 1 | 18 | 15 | 15 | 11,389,507 |
| 2 | 243 | 191 | 184 | 21,759,383 |
| 3 | 2,874 | 2,455 | 2,256 | 7,581,788 |
| 4 | 28,000 | 30,519 | 25,909 | 742,213 |
| 5 | 205,416 | 356,462 | 266,101 | 28,240 |
| 6 | 1,168,516 | 3,766,700 | 2,239,790 | 428 |
| 7 | 5,402,628 | 32,719,467 | 12,567,043 | 7 |
| 8 | 20,776,176 | 186,297,009 | 24,415,346 | – |
| 9 | 45,391,616 | 274,719,633 | 3,061,105 | – |
| 10 | 15,139,616 | 13,042,507 | 170 | – |
| 11 | 64,736 | 81 | – | – |

## H OVERESTIMATION DISTRIBUTION

In this section, we present the distribution of overestimated states for neural network models trained with both the CEA and standard CE loss functions. The distributions are shown in Figures 6–9. Each figure illustrates the number of overestimated states for both loss functions within a specific PDB. We depict the distribution by showing the number of overestimated states at each true heuristic value in the PDB, along with the range of predicted classes to which these overestimated states are assigned.

Table 4: Training hyperparameters for each PDB.

| Hyperparameter | 8-corner | 7-edge | 6-edge | $\Delta$ PDB |
|---|---|---|---|---|
| Optimizer | | Adam | | |
| Learning rate | $1 \times 10^{-3}$ | | $3 \times 10^{-3}$ | |
| Batch size | $1 \times 10^5$ | $5 \times 10^5$ | $1 \times 10^5$ | |
| $\beta$ | 1.0 | 0.6 | 0.7 | 0.9 |
| $\eta$ | 0.01 | | 0.001 | |
| ResNet blocks | 2 | | 3 | |

As the overestimation rates are also reported in Table 2, it is evident that the number of overestimated states for CEA loss is approximately $10^4 \times$ smaller than that for the CE loss across all PDBs.

## I  MODEL SIZE & HEURISTIC QUALITY

A key question is how small a model can be while still preserving admissibility and heuristic strength. We investigated this through a scaling experiment with networks of different sizes. To make the comparison fair, we kept the overall architecture—layer types and counts—the same as the network used for the 8-corner PDB in Table 2. In particular, every model keeps the initial convolutional layer unchanged, so their representation learning is identical; we vary only the number of neurons in the fully connected layers and residual blocks. We focus on the 8-corner PDB because our reference model for it already achieves nearly $100\%$ admissibility and the exact average heuristic. Table 5 represents each model's size, structure, and performance. All models were trained for the same number of iterations, although additional training would improve the smaller ones. As model size decreases, performance drops: the over-estimation rate rises monotonically and is about $10^3$ times higher in Model 5 than in Model 1. Nevertheless, Model 3 achieves almost the same performance as Model 1 while using fewer than half as many parameters, yielding a $51 \times$ compression relative to the original PDB.

Table 5: Summary of architectural details and performance for models used in the scaling experiment on the 8-corners PDB.

| Model | FC Layer Neurons | Residual Block Neurons | Size (MB) | Avg. Heuristic | Overestimation Rate |
|---|---|---|---|---|---|
| Model 1 | 1000 | 300 | 1.89 | 8.76 | $3 \times 10^{-7}$ |
| Model 2 | 800 | 250 | 1.28 | 8.76 | $3 \times 10^{-7}$ |
| Model 3 | 600 | 200 | 0.86 | 8.75 | $3 \times 10^{-7}$ |
| Model 4 | 400 | 150 | 0.51 | 8.57 | $2 \times 10^{-5}$ |
| Model 5 | 200 | 100 | 0.24 | 8.24 | $1 \times 10^{-3}$ |

## J  DYNAMICS OF $(\beta, \eta)$ DURING TRAINING

We divide the training process into multiple phases. In the first phase, we set the hyperparameters to $\beta = 1$ and $\eta = 0.1$, encouraging the model to approximate the true heuristic as closely as possible. We monitor both the loss and the overestimation rate throughout training. If the model stops making progress (i.e., neither the loss nor the overestimation rate continues to decrease), this indicates that the model cannot learn the exact heuristic across all states. At this point, we move to the next phase by gradually adjusting the hyperparameters toward more admissibility (i.e., decreasing $\beta$ and $\eta$). We repeat this refinement process until the model reaches the desired overestimation rate.

The values used for the hyperparameters $\beta$ and $\eta$ for each PDB are presented in Figure 5. For the 8-corner PDB, we did not need to adjust these parameters, as the initial values already yielded a

perfect heuristic. Our approach is to reduce both parameters whenever there is no progress in the loss or the overestimation rate. We monitored these metrics throughout training, and whenever progress stalled, we lowered both parameters to promote stronger admissibility. As a result, the number of training epochs between each reduction is not fixed. One strategy is to automatically halve both parameters every $c$ epochs when no improvement is observed. This provides a general heuristic for adapting CEA to other models and domains.

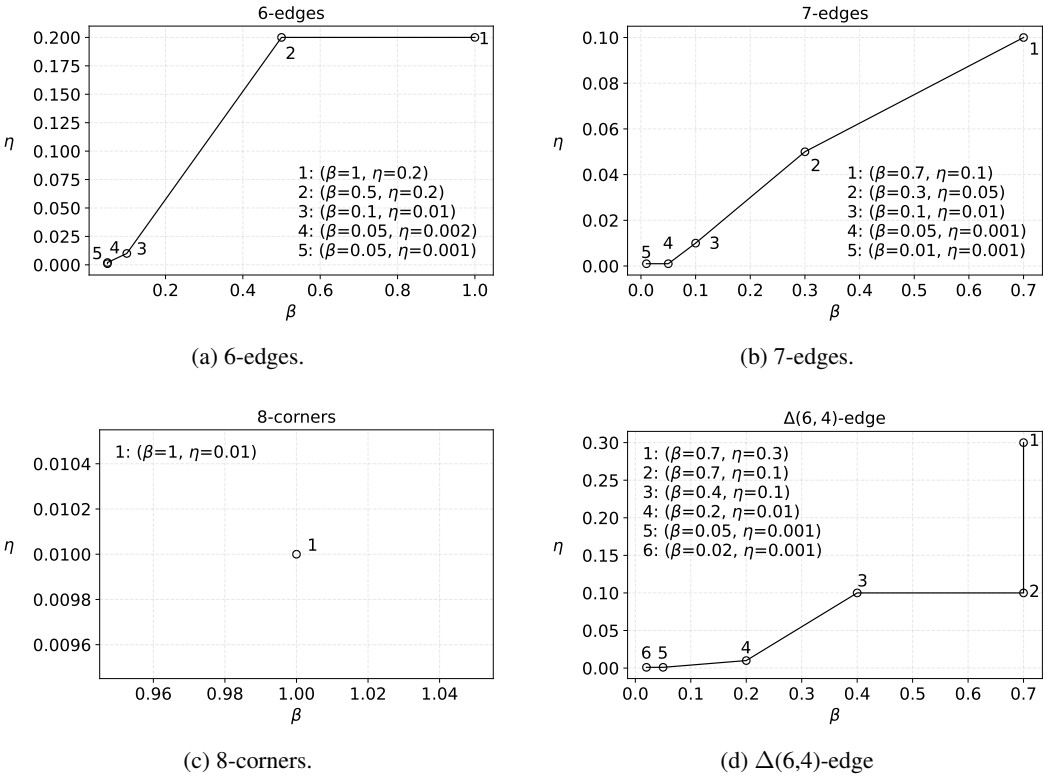

(a) 6-edges.

(b) 7-edges.

(c) 8-corners.

(d) $\Delta(6,4)$-edge

Figure 5: The $(\eta, \beta)$ values used throughout the training for each PDB.

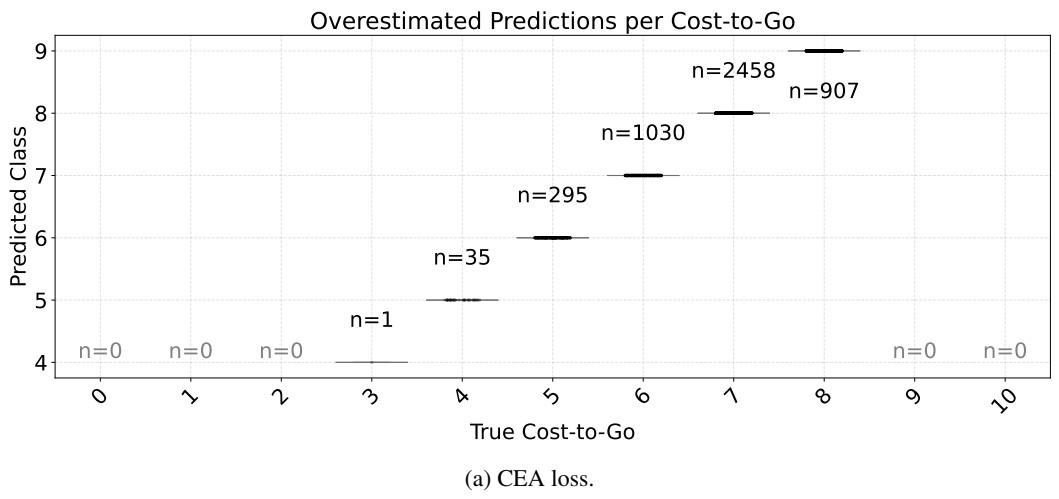

(a) CEA loss.

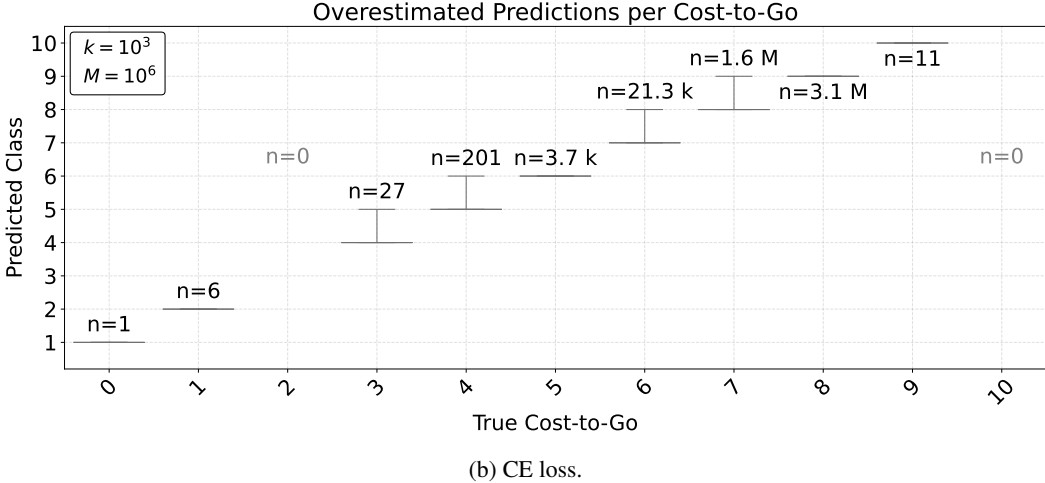

(b) CE loss.

Figure 6: Distribution of overestimated predictions per true cost-to-go for **CEA** and **CE** in **6-edges** PDB. Each box represents the spread of predicted values, while $n$ indicates the number of overestimation states for each class.

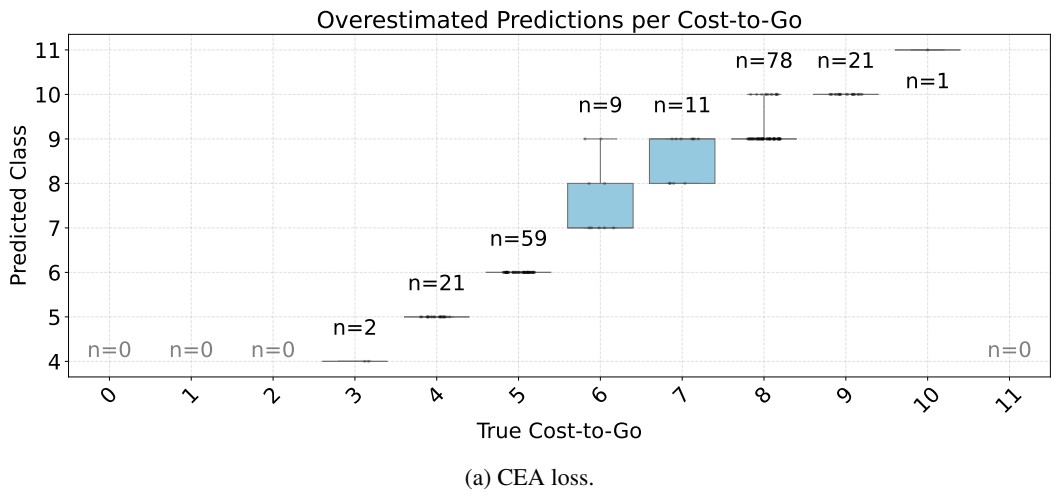

(a) CEA loss.

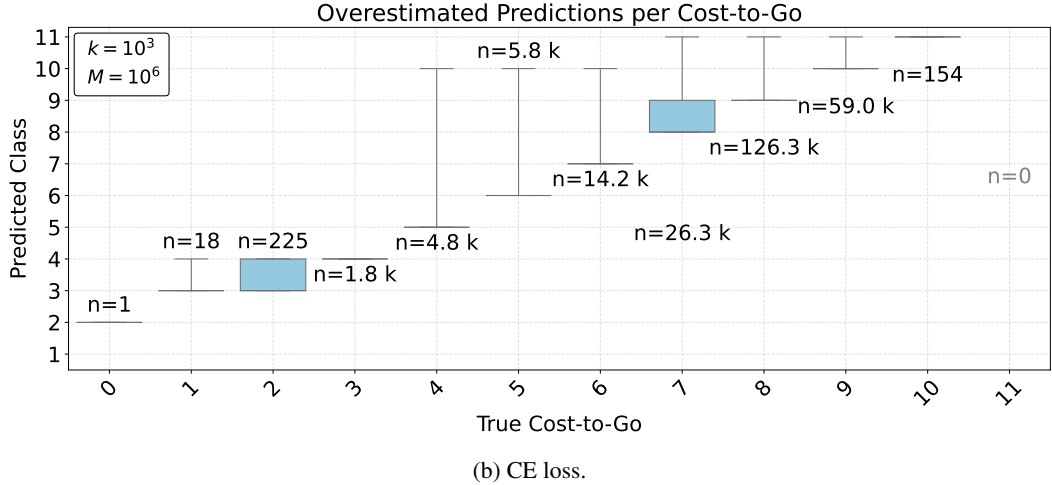

(b) CE loss.

Figure 7: Distribution of overestimated predictions per true cost-to-go for **CEA** and **CE** in **8-corners** PDB. Each box represents the spread of predicted values, while $n$ indicates the number of overestimation states for each class.

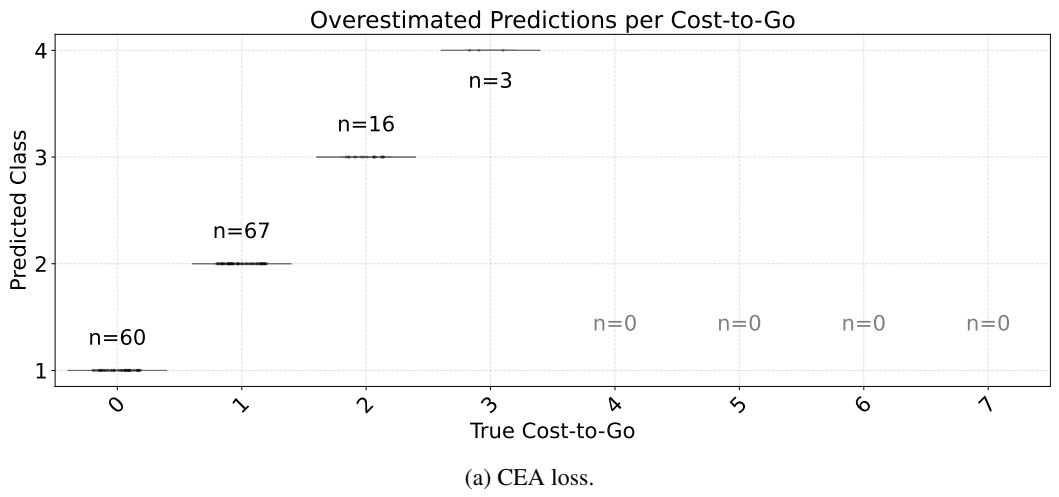

(a) CEA loss.

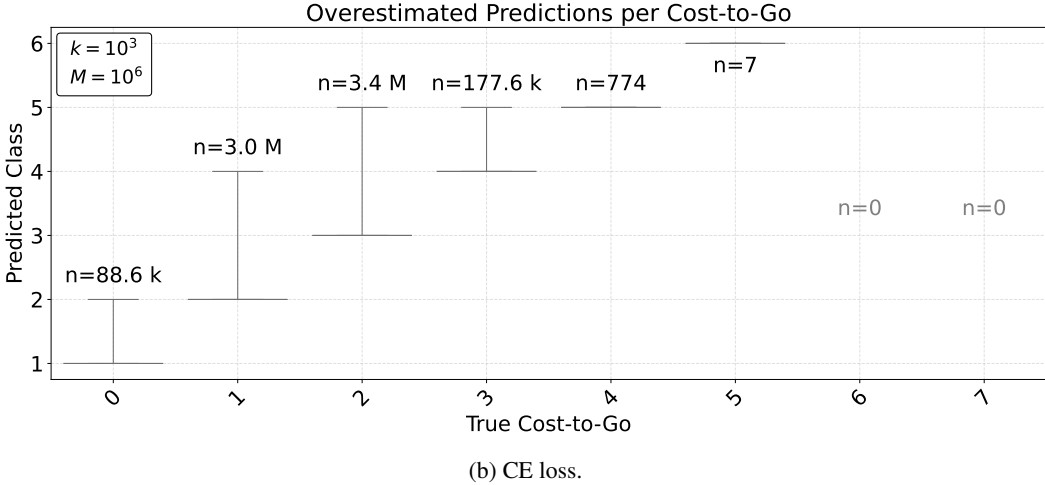

(b) CE loss.

Figure 8: Distribution of overestimated predictions per true cost-to-go for **CEA** and **CE** in $\Delta(6, 4)$**-edge** PDB. Each box represents the spread of predicted values, while $n$ indicates the number of overestimation states for each class.

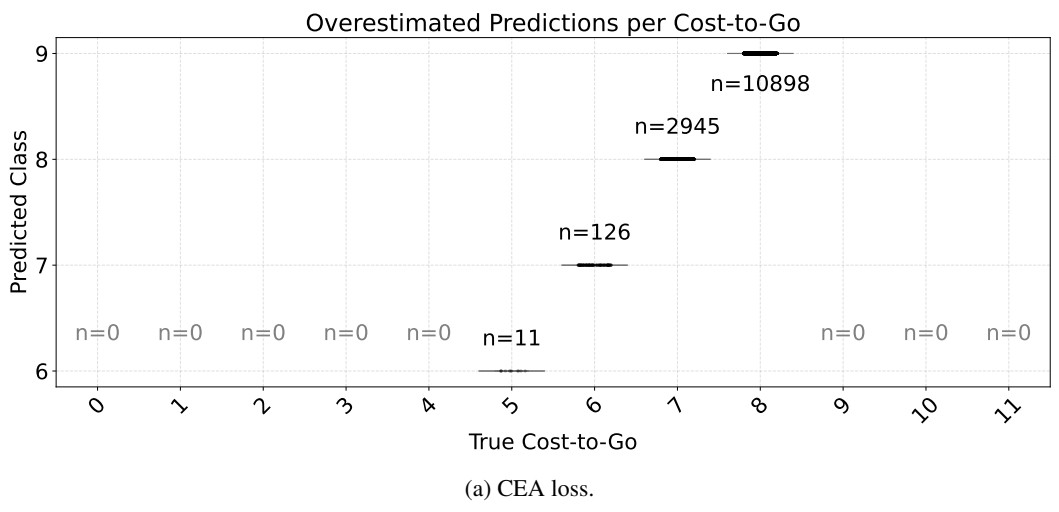

(a) CEA loss.

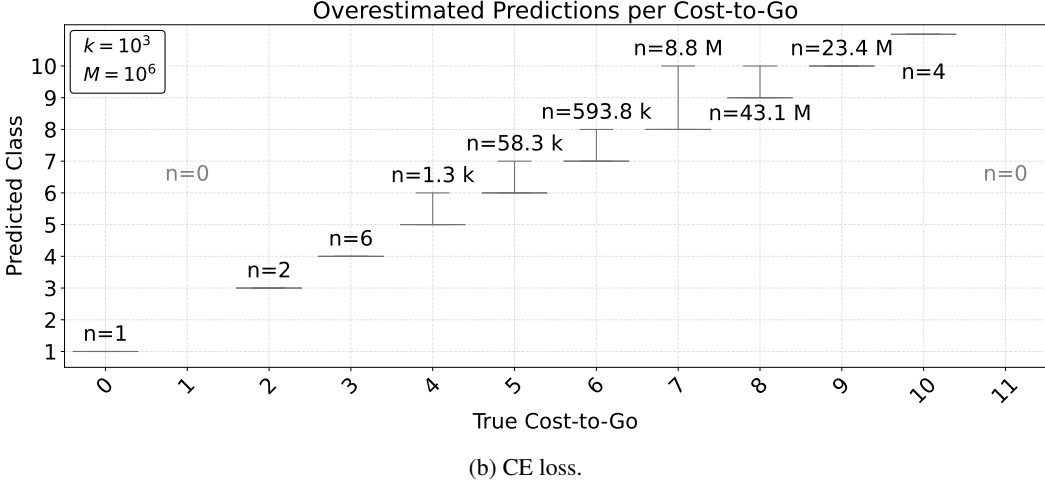

(b) CE loss.

Figure 9: Distribution of overestimated predictions per true cost-to-go for **CEA** and **CE** in **7-edge** PDB. Each box represents the spread of predicted values, while $n$ indicates the number of overestimation states for each class.

