# OpenReview forum: "Learning Admissible Heuristics for A*:  Theory and Practice"
_ICLR.cc/2026/Conference — ICLR 2026 Poster_

### Official Review · Reviewer_BWMp · 2025-10-30

**Soundness:** 2
**Presentation:** 3
**Contribution:** 2
**Rating:** 4
**Confidence:** 3

**Summary:**

The paper develops theory and practice for learning admissible A* heuristics. It tightens sample-complexity bounds under equal-cost edges and gives a sharper suboptimality guarantee that bounds A*’s error by the maximum inadmissibility on the optimal path, with extensions to neural-network and goal-dependent settings.
Practically, it introduces the Cross-Entropy Admissibility (CEA) loss, which shifts probability mass onto admissible classes and sharpens around the true class.
On Rubik’s-Cube PDBs, CEA-trained models achieve near-admissibility and match or exceed the average heuristic of same-size compressed PDBs

**Strengths:**

* The theory derives tighter generalization bounds that incorporate neural network width/depth and handle goal-dependent heuristics.
* Heuristic learning is cleanly framed as a constrained optimization problem.
* The suggested CEA loss seems to effectively encourage admissibility in the provided experimental results.
* Overall, the paper is well written.

**Weaknesses:**

* The theoretical results mostly assume equal-cost edges (Assumption 4). While this is common for toy problems like Rubik’s Cube it limits immediate transfer to domains with varied costs.
* While the CEA loss suggested in Equation 5 seems conceptually plausible, it lacks a clear motivation why this specific function was chosen. The goal of shifting some probability mass towards admissibility could be expressed through other loss functions as well, for example by minimizing the kl divergence between the predictions and a suitably defined target distribution. Were other loss functions considered? If so, why was this specific one chosen?
* Some details of the experimental results are vague. No details on how the data is split into train/validation/test splits is provided.
* As the theory in section 3 derives the sample complexity for learning heuristics, an ablation study that measures the heuristics performance as a function of the training set size would strengthen the connection between the works theoretical and empirical sections.
* The CEA loss introduces β and η as additional hyperparameters that are adjusted in stages during training, which adds complexity to the training process.

While the theoretical results and novel CEA loss are contributions, I do think these weaknesses currently outweigh the strengths of the paper.

**Questions:**

* How was the data split for the empirical study? How many samples are used for training?
* Can the CEA loss be extended to settings with non-uniform edge weights?

---

> ### Author Response · Authors · 2025-11-19
> **Response to Reviewer BWMp (Weaknesses)**
>
> Dear Reviewer BWMp,
>
> We appreciate your time on this review.
>
> ### Weaknesses
>
> **Assuming equal-cost edges (Assumption 4).**
> There is no admissible learned heuristic currently available for Rubik’s Cube. Moreover, the equal edge-cost condition holds in many planning domains. In addition, many graphs with positive edge costs can be converted to an unweighted graph via a standard reduction, which increases the number of states but preserves equal edge costs. Here are the two main reasons we include Assumption 4 in parts of our analysis:
>
> 1. Bounds on general graphs with a general performance measure are already established in [2]. Our goal was to improve these bounds for a specific class of graphs that satisfy Assumption 4.
> 2. To the best of our knowledge, PDB heuristics are available for graphs that satisfy Assumption 4. Since our goal is to use PDB heuristics to learn admissible heuristics, we incorporated this assumption into our theorems, which also tightened the bounds.
>
> For the upper bounds on expected suboptimality, we do **not** use Assumption 4; those results apply to all graphs. Likewise, neither Theorem 6 nor Theorem 7 requires Assumption 4 to derive the bounds. We emphasize this assumption because our experiments focus on such graphs. For a more general class of graphs, the bounds in these theorems can be obtained by using $\mathcal{O}(n^2 \log n)$ instead of $\mathcal{O}(n \log n)$; we omitted these variants due to space constraints.
>
> **Motivation behind CEA loss.**
> Our goal was to design a loss function that pursues a near-perfect heuristic while still favoring underestimation. We first experimented with mean-squared error and treated the problem as regression, but admissibility was extremely poor (around 30%). Even when predictions were close, small positive errors broke admissibility (e.g., $h^*(s)=6$ and $h(s)=6.012$).
>
> We then tried cross-entropy and treated the problem as classification, using label smoothing to distribute probability mass over lower classes (e.g., for $h^*(s)=5$, a target such as `[0.005, 0.015, 0.02, 0.06, 0.9]`). This also performed poorly because even admissible predictions incurred non-zero loss if the probability distribution over other classes did not match the target. We also tried KL divergence; while KL works well in policy-gradient methods to keep updated policies close to previous ones, in our setting it is unclear how to define a suitable target distribution for each state. These limitations motivated the design of the CEA loss.
>
> **Tuning η and β.**
> Please see Section J for details. In summary, we do *not* tune these values. We start with initial values for both hyperparameters and periodically reduce them if the desired admissibility is not achieved after a specified number of epochs. Figure 4 reports all values used during training. A simple automated schedule—used in the Table 5 experiments—is to halve both values when admissibility stops improving. Intuitively, reducing these parameters indicates that the model cannot learn an admissible heuristic at the current strength, so we encourage more underestimation.
>
> **Connection between theoretical and empirical sections.**
> We address this in our response to the corresponding question in the Questions section.

---

> ### Author Response · Authors · 2025-11-19
> **Response to Reviewer BWMp (Questions)**
>
> ### Questions
>
> **Data split for the empirical study.**
> This is a great point. Let us first clarify one important distinction. Our setting differs from standard supervised learning. In typical supervised learning, training and validation sets are used to generalize to a test set that represents a much larger unseen population. In our case, each PDB contains *all* abstract states of a graph, and we have a ranking function mapping every Rubik’s Cube state uniquely to a PDB entry. Therefore, if a model predicts an admissible heuristic for every PDB state, we obtain a fully admissible heuristic for Rubik’s Cube. Larger PDBs use weaker abstractions and retain more information, which is why they contain more states.
>
> To your question: we have millions of states in each PDB and a very small model for fast inference (our 8-corner model is $30\times$ smaller than DeepCube [1] while delivering an admissible heuristic). The goal is to compress the information in these large datasets into a tiny model while losing as little information as possible. To this end, we sample uniformly random training batches (we address data imbalance by subsampling) from the entire set for a fixed number of epochs. Inevitably, many individual states will never be seen during training, but the sampling covers the full distribution. As a result, our models achieve very low inadmissibility rates while preserving far more information than traditional compression techniques. In future work, we aim to compress much larger PDBs such as the 8-edge PDB with **5 billion** states (2.4 GB), though this is beyond our present scope.
>
> **Can the CEA loss be extended to settings with non-uniform edge weights?**
> Yes. CEA does not impose constraints on edge-cost definitions; it depends only on heuristic values. If the domain admits a discrete (finite) set of unique heuristic values, CEA can be applied even with non-uniform edge weights.
>
> [1] Solving the Rubik’s Cube Without Human Knowledge.
> [2] Sample Complexity of Learning Heuristic Functions for Greedy Best-First and A* Search.

---

> > ### Comment · Reviewer_BWMp · 2025-11-27
> >
> > I thank the authors for the clarifications and improvements to the manuscript.
> >
> > While the narrow focus on the Rubik's cube domain remains a relative weakness, my main concerns have been addressed and I am raising my score accordingly.

---

### Official Review · Reviewer_ntxk · 2025-11-01

**Soundness:** 2
**Presentation:** 3
**Contribution:** 3
**Rating:** 6
**Confidence:** 4

**Summary:**

This paper extends existing theoretical analyses on the sample complexity of
learning admissible heuristics, then proceeds to define a new loss function
to achieve such admissibility.
The contributions in this work are

-   Thm2: Extension of Sakaue & Oki 2022's result on Pdim to PDB's abstracted states
-   Thm3: a tighter suboptimality bound when A\* with is allowed to reopen nodes &#x2013; extension of Valenzano '14,
-   Thm4: Eq. (3) which has O(sqrt(n)) instead of O(n) in Eq (1) = Prop 1 + Sakaue & Oki 2022.
-   Thm5: Extension of Sakaue & Oki 2022's result on Pdim to NNs, with/without PDBs
-   Thm6: Extension of Thm5 to NNs with specific sizes
-   Thm7: Extension of Thm6 to different goals
-   proposing CEA loss.

I have a mixed impression about this work. While the paper is tackling an important problem,
the paper appears to be written by two authors without a necessary organic integration,
which is a common phenomenon &#x2014; one symbolic author and one machine learning author failing to understand each other, stuck in their own worldview.
This might be causing inconsistent math notations (e.g. log vs. lg, $\ell$ vs. $D$),
overlapping symbols ($W$ as cost upper bound vs parameter size),
and an apparent disconnect between the theoretical result and the empirical result described below.

The paper did excellent work coverint the backgrond on the sample complexity analysis.
However it is light on proofs, which are largely deferred to the appendix,
and has several minor notation issues which must be fixed.

I also have a bigger concern about the practical part of this paper: CEA, and the empirical evaluation.

**Strengths:**

The paper did excellent work covering the backgrond on the sample complexity analysis.

**Weaknesses:**

## Issue 1: CEA

As discussed in [Nunez-Molina 2024] cited in this work,
the choice of the loss function must directly correspond to the statistical representation of the output.
Categorical target random variable yields the cross-entropy loss naturally as a negative log likelihood.
The proposed CEA loss does not have a corresponding distribution.
If the author wishes to impose some mathematical property on the prediction target,
the right way to design a machine learning model is to idenfity the correct representation of the target as a random variable,
then derive the loss function naturally from its pdf.

One such alternative as a baseline for comparison against CEA I suggest is **ordinal logistic regression** for ordinal variables.
Ordinal regression [1] is a classical method for predicting ordinal random variables,
which naturally address the lack of orders in the categorical prediction (classification).
If the model believes that the output is larger than k,
then the model architecture must hard-code the invariance that the output is also larger than k-1, k-2, and so on.

[1] McCullagh, Peter (1980). "Regression Models for Ordinal Data"

One of its simplest implementation uses the cumulative binary logistic loss,
which outputs a single scalar with a NN, subtract monotinically increasing threshold values,
apply a sigmoid, then train the output with binary cross entropy.

parameters:

-   A scalar output neural network $f$ (the last layer does not need the bias)
-   A bias vector $\mathbf{b} \in R^k$ for prediction over $\{0, 1, ... k\}$

how:

-   input $x$
-   target value $h^* \in ｛0, 1, ... k｝$
-   binary prediction target $\mathbf{y} \in ｛0,1｝^k$:
    -   $y_h = 1$ if $h > h^*$ ;
    -   $y_h = 0.5$ if $h=h^*$ ;
    -  0 otherwise.
    -  For example, it looks like $[0,0,0,0,0.5,1,1,1]$ when $h^*=4$ and $k=7$.
-   scalar logit $l = f(x)$
-   thresholds $\mathbf{\theta}$
    -   $\theta_0 = b_0$
    -   $\theta_h = \theta_{h-1} + softplus(b_h)$
        -   This ensures $\theta_h > \theta_{h-1}$
-   output: vector $\mathbf{p}\in [0,1]^k$, where
    -   $p_h = \sigma( l - \theta_h )$
    -   $p_h$ represents $p(h^* < h | x)$ , and this framework guarantees $p(h^* < h | x) < p(h^* < h+1 | x)$ .
-   loss: $\sum_h CE( y_h, p_h )$

Notice that the final loss naturally penalizes the misprediction of $p(h^* < h | x)$ the further $h$ is away from $h^*$.

For example, when $h^*=4$ and $p_6$ is smaller than 0.5, the framework ensures that $p_4, p_5$ are even smaller, resulting in three times or more penalty.
This has a similar characteristics as the first term of the CEA, while maintaining the natural statistical interpretation.

You may however return a rebuttal: Then how do we interpret the tunable &beta; in CEA, and the fact that it only applies to the overestimation?

My answer: As cumulative binary logistic loss is a sum of losses for a series of binary predictions,
we can achieve the same effect by multiplying a set of coefficient to each loss.
This is a common practice for addressing a class imbalance (although our purpose is different)
and it is mathematically equivalent to oversampling/undersampling a subset of the dataset.
(Well, there may indeed be a class imbalance.
see the next section blow.)
Concretely, use the loss as follows:

-   loss: $\sum_h w_h CE( y_h, p_h )$

You are free to choose any $w_h$, e.g. $w_h = g(h, h^*)$ for some formula g that increases as $h$ overestimates.
So, there is no ad-hoc tricks; It simply artificially increases the penalty to the overestimation.


## Issue 2: PDB learning

I found that the method section and the evaluation section is quite ambiguous about the training data.
I struggled to discover what is the input representation to the NN, what is the taget value, and how many data points are used.
Please add a separate section titled "data collection" which details what goes in the pipeline.
I initially had 4x2 = 8 possible interpretations:

-   Input x to the NN:
    -   Raw state representation, factored
    -   Raw state representation, in a form of state id
    -   PDB abstract state representation, factored (\*)
    -   PDB abstract state representation, abstract state id
-   Target data:
    -   optimal solution cost $h^*(x)$ obtained by search from x using admissible PDB
    -   admissible heuristic $h(x)$ from the PDB &#x2014; which is an optimal path cost in the abstract state space
-   The number of data points

The detail of the input representation is only given in the appendix.
While the full detail is not necessary, the main paper must at least mention
that it is the factored abstract state representation of PDB.

What's more important is the target data.
I know you used $h^*$ in Eq.5, but I don't have a good confidence because
there is no mention of how it was collected.

I could not find any information about the number of data points.
Did you use the entire abstract states in the PDB without subsampling?
If so, the dataset is hugely imbalanced, as you pointed out in the paper.
The standard practice is oversampling, undersampling, and bagging. See [1].
These standard measures completely normalizes the number of samples per class.

The lack of mention to how the author addressed the class imbalance imply a danger that
the perceived improvement from the CEA loss could be just an artifact of **accidentally** somewhat addressing
the class imbalance in an ad-hoc way.
(As said above, multiplying $(k/h^*)^\beta$ to the loss is mathematically equivalent to oversampling.)

The authors also mentioned delta-PDB, which would shift h's distribution and also somewhat reduces the class imbalance,
but the Table 3 shows that the entire dataset still contains a massive class imbalance.

[1] Wallace, Byron C., et al. "Class imbalance, redux." 2011 IEEE 11th international conference on data mining. Ieee, 2011.

## Issue 3: Disconnect between theoretical results vs experiments

The theoretical and the empirical results in this paper are, while individually interesting, quite disconnected.


### theorem 4

First, the paper does not empirically verify Theorem 4.
I expected an experiment that estimates the LHS of eq.3 using a validation set as a surrogate or the entire (abstract) state space,
subtract the first term of Eq.3, then plot the remaining value to confirm that it grows $O(\sqrt(n))$.

Theroem 4 does not require factored state representation, Rubiks cube, PDBs or the CEA loss function.
Instead, theorem 4 is a function of the graph size n and the sample size N.

In other words, while section 3.1 is by itself an interesting contribution,
its empirical confirmation must be done separately using a state space whose size is configuratble,
e.g., not in Rubik cube, but in artificial random graphs such as those studied by Cohen & Beck [AAAI17, AAAI18, IJCAI18]
whose $n$ can be configured.

-   Problem Difficulty and the Phase Transition in Heuristic Search, AAAI17
-   Local Minima, Heavy Tails, and Search Effort for GBFS, AAAI18
-   Fat- and Heavy-Tailed Behavior in Satisficing Planning, IJCAI18


### theorem 7

Another main theoretical results in this paper is the sample complexity in theorem 7.
I expected that the empirical analysis verifies this by one of these experiments:

-   Experiment 1: Given a fixed network, compute this upper bound Pdim(U) with
    Thm7. Then gradually reduce the dataset size by subsampling. See what happens
    if the dataset size goes below Pdim(U)
-   Experiment 2: Given a fixed dataset, reduce the number of weights in NN to reduce Pdim(U).
    See what happens if Pdim(U) goes below the dataset size.


## Issue 4: Runtime analysis of using the NN

While the paper claims that it tries to find a smaller NN (Eq.4),
it does not do anything about it actually.
The evaluation focuses on accuracy and admissibility,
and never compares the actual runtime or the number of node evaluations
between the sota solver against A\* guided by the heuristics learned by this method.

In other words, as is often the case with many machine learning papers,
I suspect that this paper obtained a quite accurate heuristic that is too slow
to compute to use in practice.


## Minor comments and questions

-   line 80 : b should be a neighbor of a?
-   l.77-91: floor and indicator functions are not used in the paper?
-   l.165-173: better to have a consistency in the choice of letters. H -> U?
-   l.173: h suddenly becomes a real vector. I know this is an array from a state to R, but this is inconsistent to the previous notation h(s) of state s.
-   l.213: $|G_v| <= n^2$ I suppose.
-   l.215: same regarding h.
-   l.215: add "derived from S & Oki 2022".
-   l.216: same reagarding h.
-   l.234: "general class of performance measure", provide examples, e.g., runtime, memory, node evals, etc.
-   l.252: Results in Valenzano 14 holds regardless of admissibility, which is surprising.
    Although it is not wrong not to mention it (because nothing more is not assumed than stated),
    due to the strong association between A\* and admissibility,
    I strongly suggest the authors to explicitly state that the entire section 3.1 does not depend on the admissibility of h.
-   l.268: $\hat{H}$ undefined.
-   l.270: lg n vs log n.
-   l.302: sum of $w_1 \ldots w_n$ is denoted by U. U -> W?
-   l.305: Indeed, here the number of parameters is W.
-   l.317: the number of output classes is $\ell$. It was $H$ before. or $\hat{H}$. wtf?
-   l.326: this time the number of output classes is $D$. wtf?
-   l.333: Rep(I) is not used at all elsewhere in the paper. maybe appendix. please remove it or move it to the appendix, if used there.
-   l.343: $\ell$ and $D$ used at the same time. Also, U and W are used at the same time. $\ell'$ undefined.
-   l.364: $|h|$ is never explicitly optimized. I doubt Eq.4 is necessary.

**Questions:**

Q1. Confirm the input format is PDB abstract state representation, factored. (yes/no)

Q2. Confirm the target value is really h\* and is obtained by solving x optimally. (yes/no)

Q3. Did you use the entire abstract states in the PDB without subsampling? (yes/no)

Q4: The author claims that Eq.(3) is tighter than Eq.(1) with no explanation.
I assumed O(sqrt(n)) vs O(n) is what they implied. Am I correct? (yes/no)

Also, please add a sentence that explains conceptually why this reduction is achieved.
(E.g., in sorting literature, radix sort can run faster than O(n log n) &#x2014; why?
it is because it can use more information than comparison sort.)
I assume that this is in line with something like: it uses the max in Thm3,
therefore only one element along the path affects the error,
while the first error term in Eq.(1) is essentially a double summation (sum over paths, sub over states on the path)

---

> ### Author Response · Authors · 2025-11-19
> **Response to Reviewer ntxk (Weaknesses) - Part 1**
>
> Dear Reviewer ntxk,
>
> We appreciate your time on this review.
>
> ### Weaknesses
>
> The misalignment in the notations mostly stems from similarities between the notation used in ML theory and classical planning. We will address all inconsistencies in the revised version.
>
> **Issue 1: CEA.**
> Ordinal regression is the setting in which we frame our problem, and the first loss function we explored was the cumulative binary logistic loss. The benefit of this loss is that by learning thresholds, class probabilities are ordered by construction. However, the approach you suggested, using a loss of the form $\sum_h w_h * CE(y_h, p_h)$,
> does not provide the same effect as our tunable $\beta$ and $\eta$
> parameters, which explicitly encourage admissibility. Let us explain:
>
> Because $y_h$
> is a one-hot vector, for a state with true class $h^{*}(s)=c$
> , the loss becomes:
>
> $
> L = \sum_{h=1}^{K} w_h\ CE(y_h,p_h)
>    = -\sum_{h=1}^{K} w_h\ y_h \log p_h
>    = -w_c \log p_c,
> $
>
> This is exactly the weighted CE loss used for class imbalance. Since $w_h$
> is constant for all states sharing the same true class h*, all states in that class receive the same multiplier, even though their predictions may differ greatly. For example, if h*(s1)=h*(s2)=5 and the predicted softmax vectors are:
>
> $p(s1) = [0.10, 0.20, 0.39, 0.01, 0.30, 0, 0] $
>
> $p(s2) = [0, 0, 0, 0, 0.30, 0.70]$
>
> then these two states should not receive the same loss. With the proposed loss, both are scaled by the same $w_h$
> . To favor predictions like p(s1), where most probability mass lies on lower classes, the loss must penalize at the state level. Our CEA loss encourages placing more mass on lower classes (h < h*) without enforcing a strict total order. As beta decreases, all admissible classes are treated more similarly. CEA also shares the smoothness and gradient-flow properties of CE, has a single global optimum at the true class, and treats other admissible predictions as local optima.
>
> Additionally, datasets may contain heavily populated high-heuristic classes (e.g., 8 and 9 in the 7-edge PDB). A constant multiplier emphasizing these classes can hurt admissibility for lower classes. We address imbalance using the sampling method described later.
>
> **PDB learning.**
> We address the grouped concerns one by one.
>
> 1) **Training details.**
> Experimental details were moved to the appendix due to space limits. Section G explains the input representation for each PDB. In short, inputs encode the abstract state. For example, in the 8-corner PDB, each face is represented using six 3x3 channels, for a total of 36 channels. PDBs store the exact h* value for each abstract state. During search, every state in the main graph maps uniquely to one abstract state, and we use that h* as the training target. No data collection is needed because PDBs are computed once via optimal search and are publicly available. See references [1] and [2].
>
> 2) **Data imbalance.**
> PDB classes are highly imbalanced. We first tried focal loss [3], but it did not help. Instead, we correct the imbalance at the sampling stage: each batch samples uniformly within each heuristic class, and the number of samples per class is proportional to its size. Without this sampling method, results were far worse. Section I of the appendix also studies model-size vs. heuristic-quality trade-offs under multiple random seeds. Since we consistently obtained similar admissibility rates across all settings (random seeds and model sizes), these results cannot be attributed to chance.
>
> 3) **Delta-PDB.**
> We omitted many delta-PDB details for space. Delta-PDBs have been used to reduce the memory of PDB heuristics and increase the potential compression [8; Section 6.1]. This concentrates the mass on fewer (lower) classes. Our aim was to test learnability in this setting: although class imbalance persists, the mass is concentrated on fewer, lower classes, which helps learning when encouraging underestimation for admissibility. Compare an extreme imbalance in favor of class 8 versus a extreme imbalance in class 2: overfitting in the first case will likely induce widespread inadmissibility (for states with heuristic $<8$), whereas in the second case inadmissibility impacts far fewer classes.

---

> > ### Comment · Reviewer_ntxk · 2025-11-27
> > **Thanks for the reply**
> >
> > Thank you for the reply and for the addition of sampling details in the revised paper.
> >
> > > Since $w_h$ is constant for all states sharing the same true class h*,
> >
> > This argument is a straw-man because I did not say $w_h$ is a constant with respect to $h$.
> > In the last sentence of Issue 1, I stated that $w_h$ can be a function of $h$ and $h^{\ast}$.
> > For example, why not $w_h = \mathbf{1} [[h > h^\ast]] (h-h^\ast)^2$ ?
> >
> > My main complaint is the lack of mathematically clean justification for the CEA loss. If you still believe the CEA loss is superior, why not  instead reformulate CEA loss in a way that can be reduced to the standard ordinal regression?
> >
> > > CEA also shares the smoothness and gradient-flow properties of CE, has a single global optimum at the true class, and treats other admissible predictions as local optima.
> >
> > Although trivial, I do not remember this mentioned in the paper, so it is a good idea to add it. By the way, I assume the $w_h$ definition above is also smooth wrto $h$ and has the optima at $h=h^\ast$, so it is not a good rebuttal.
> >
> >
> >
> > Btw, a totally unrelated comment for everyone's interest: I found \ast can safely typeset $\ast$ in MathJax without messing up the markdown processor.

---

> > > ### Author Response · Authors · 2025-11-30
> > > **Re: Assumption of a constant w_h**
> > >
> > > It seems our earlier explanation was not sufficiently clear. When we stated that $w_h$ is a constant for all states sharing the same true class $h^\*$, we did not mean it is a single scalar value. Rather, we meant that the **vector** $w_h$ is the same for all states with true class $h^\*$, regardless of their predicted probabilities. For example, in the scenario you mentioned — *$w_h = \mathbf{1}[h > h^*](h - h^*)^2$* and *$L = w_h \sum_h CE(y_h, p_h)$ — if $h^\* = 5$* and there are eight classes in total, then every state with true class 5 will share the same weight vector *$w_h = [0,0,0,0,0,1,4,9].$*
> > >
> > > This behaviour is expected: incorporating the predicted class $h$ directly into the loss would require an **$\arg\max$ operation**, which would break differentiability. In CEA, we similarly obtain a constant weight vector for each $h^*$. The source of confusion appears to be our earlier assumption that $y_h$ is a one-hot vector. As you noted in your first comment, **this is not the case**, and we overlooked this detail. We elaborate further on the distinction between CEA and standard ordinal regression in the following comment.

---

> > > ### Author Response · Authors · 2025-11-30
> > > **Re: Mathematical Justification of CEA**
> > >
> > > Thank you for raising this point. We did not initially include a formal mathematical justification for CEA, as it is primarily built upon cross-entropy (CE) but modified to favor underestimation. Your question gives us the opportunity to present this justification in more detail. We highlight two aspects:
> > >
> > > ### **1. Statistical behavior of the loss**
> > >
> > > The CEA loss can be viewed as the negative log of a utility function that rewards probability mass on the true or underestimated classes. For a given example with label $h^*$, we define
> > >
> > > *$U(p; h^\*) = \sum_{k=1}^{h^\*} \Bigl(\tfrac{k}{h^\*}\Bigr)^\beta  p_k,$*
> > >
> > > which aggregates the model's predictive probability over all classes $k \le h^\*$, with larger weights for classes closer to $h^\*$. The first term in CEA is then $-\log U(p; h^\*)$, a log-loss that favors distributions concentrating mass on “safe’’ (true or underestimated) classes. The second term, $\eta[-\log p_{h^*}]$, is the standard cross-entropy on the true class. Thus, CEA is a smooth, differentiable loss that combines the usual negative log-likelihood with an additional term that explicitly encourages underestimation.
> > >
> > > CEA is also **convex**. For a fixed label $h^*$, the first term in CEA can be written as
> > >
> > > *$
> > > \ell_1(p; h^\*)=
> > > -\log \Biggl(
> > >   \sum_{k=1}^{h^{\*}}
> > >     \Bigl(\tfrac{k}{h^{\*}}\Bigr)^{\beta}
> > >     p_k
> > > \Biggr)=f\bigl(g(p)\bigr),$*
> > >
> > > where $p = (p_1,\dots,p_K)$, $w_k = (k/h^\*)^\beta > 0$ for $k \le h^\*$, and
> > >
> > > *$g(p) = \sum_{k=1}^{h^\*} w_k p_k,\qquad f(x) = -\log x.$*
> > >
> > > The map $g$ is linear in $p$, hence both convex and concave. The function $f(x)=-\log x$ is convex on $(0,\infty)$ and strictly decreasing. By the standard composition rule for convex functions, $f \circ g$ is convex in $p$. The second term in CEA is the standard CE, also convex in $p_{h^*}$. Since the first term is convex in $p$ and $\eta \ge 0$, the entire per-example loss is a nonnegative linear combination of convex functions and therefore **convex in $p$**.
> > >
> > > ### **2. Proper scoring and consistency**
> > >
> > > We also require that the true distribution be the unique minimizer of the expected loss. Here, the true distribution places all probability mass on the true class $h^*$. For such a distribution, both terms in CEA are minimized (and equal to $0$) when the model prediction $p$ also places all probability mass on the true class. Any other admissible distribution incurs a strictly higher loss. Thus, the global optimum is achieved precisely when the entire probability mass is assigned to the true class.

---

> > > ### Author Response · Authors · 2025-11-30
> > > **Re: CEA vs. standard ordinal regression**
> > >
> > > We do not claim empirical or theoretical superiority over standard ordinal-regression losses such as the cumulative binary logistic loss. Our objective is different: to adapt the cross-entropy loss so that it aligns with the requirements of learned heuristics for classical search algorithms.
> > >
> > > The key difference between CEA and cumulative binary logistic loss lies in how class orderings are encoded. In cumulative binary logistic loss, class order is hard-coded into both the model and the loss through learned thresholds, guaranteeing monotonic cumulative probabilities. In contrast, CEA imposes ordering only through the loss function, using weighting parameters and by ignoring probability mass on overestimated classes. There is no architectural constraint enforcing monotonicity; instead, ordering is imposed softly among admissible classes.
> > >
> > > Our motivation for this design is practical: in very large graphs with millions of states, enforcing strict monotonic constraints across all classes is challenging, whereas soft constraints make it feasible to obtain (nearly) fully admissible heuristics across the entire graph. **That said, this is a design choice rather than a proven advantage.** Determining whether the soft constraints in CEA consistently help or even outperform the cumulative binary logistic loss on very large datasets would require further experimentation, which we consider beyond the scope of the current work.

---

> ### Author Response · Authors · 2025-11-19
> **Response to Reviewer ntxk (Weaknesses) - Part 2**
>
> **Disconnect between theoretical results and experiments.**
> We address these concerns per theorem:
>
> **1) Theorem 4.**
> Theorem 4 tightens the upper bound on suboptimality by allowing A* to reopen nodes. Here, the heuristic is in $R^{n}$ (no function approximation), so there is no state representation or CE loss in Eq. (3). Regarding PDBs: a PDB reduces the main graph to an abstract graph where $n$ becomes $m$; the theorems and bounds derived from this reduction are straightforward but omitted for space. Since we did not improve the generalization bound on *expected* suboptimality, we did not further pursue those experiments; illustrative examples are provided in [4].
>
> **2) Theorem 7.**
> Theorem 7 treats goal-dependent heuristics (any state may be a goal). Unfortunately, no goal-dependent PDB dataset exists with $h^{*}$ for every goal; constructing one would be enormous. For instance, our smallest PDB has 42M states; the corresponding goal-dependent dataset would have
>
> $1.76\times 10^{15}$
> states.
>
> If you meant the experiment verifying Theorem 6, Table 2 satisfies the bound on the number of training instances. We also conducted the experiment you refer to as **Experiment 2** (Section J), fixing the PDB (8-corner) and shrinking the model to explore how far compression can go while retaining heuristic quality. Please see Table 5 in that section.
>
> ---
>
> **Runtime analysis of using the NN.**
> We answer in two parts.
>
> **1) Model size.**
> We chose model sizes to be small enough to yield strong compression relative to the PDB memory. We further optimized the 8-corner model size in Section J. These models are designed for fast inference. Notably, our 8-corner model is $30\times$ smaller than DeepCube [5] while delivering an admissible heuristic.
>
> **2) Heuristic strength.**
> The source PDBs are not weak; for example, 7-edge and 8-corner have strong average heuristics. From the learned distributions (Figures 5–8) and averages, models trained with CEA loss retain most information, so search performance matches PDBs closely. Current classical search algorithms are not yet engineered to exploit NN heuristics efficiently; end-to-end runtime is thus an algorithmic issue. We study the design of efficient search algorithms in our recently accepted AAAI paper.

---

> > ### Comment · Reviewer_ntxk · 2025-11-27
> > **reply**
> >
> > > Theorem 4
> >
> > I reiterate that we have not seen $O(\sqrt{\frac{n+\log \frac{1}{\delta}}{N}})$ experimentally.
> >
> > > Theorem 7
> > > Model size
> >
> > Indeed, Theorem 6 would be enough. Thanks for pointing me to Table 5 (revised version). This at least provides an estimate of Pdim. Is Pdim not analytically computable?
> >
> > > 2) Heuristic strength.
> >
> > I still wish the authors will add runtime-based experiments in the camera-ready. Also, congrats on AAAI acceptance, hope to read it when they made the accepted papers public. You can also run the experiment with your new method, not A*.

---

> > > ### Author Response · Authors · 2025-11-30
> > > **RE: Heuristic strength**
> > >
> > > Dear Reviewer,
> > >
> > > Thanks for the comment and your message. In the camera-ready version, we will cite both the published version and the preprint version of the AAAI accepted paper (including the additional experiments in the appendix), together with a clear explanation of the results.
> > >
> > > Here is also a brief sumamry of the runtime results: in that paper, we argue that classical search algorithms are not **efficient** when used with ML-based heuristics. We compared node expansions and node generations under two settings:
> > >
> > > 1. our algorithm using ML heuristics
> > > 2. a classical search algorithm using PDB heuristics.
> > >
> > > In our results, both the number of nodes generated and the number of nodes expanded were less than half of those for the PDB heuristics, even though both heuristics have the same memory footprint. This shows that the learned heuristic is **STRONGER** than the PDB heuristic. Regarding runtime, our algorithm running on four GPUs achieved performance very close to one of the state-of-the-art parallel search algorithms.

---

> > > ### Author Response · Authors · 2025-11-30
> > > **Re: Experiments for Theorem 4**
> > >
> > > Thank you for raising this point. As mentioned in our response, we will release log files containing the admissibility error term at different stages of training for Theorem 4. However, it is indeed useful to include an explicit experiment illustrating how the generalization error behaves as the number of training instances increases. To this end, we have added a new experiment to the main paper that tracks this behaviour. Figure 2 (page 10) presents the results for the 8-corner PDB. The generalization error decreases as the number of training instances $N$ grows, following a **sublinear decay** consistent with the theoretical rate  *$\tilde{\mathcal{O}}\left(\hat{H}\sqrt{(n + \log(1/\delta))/N}\right)$*.  This empirically confirms the expected **$1/\sqrt{N}$ trend** predicted by our sample-complexity analysis.

---

> > > ### Author Response · Authors · 2025-11-30
> > > **Re: Is Pdim not analytically computable?**
> > >
> > > This is an excellent question. To address it, we briefly outline the proof strategy used in our neural network results. The key idea is to partition the neural network parameter space into regions within which the network’s output can be expressed as a **polynomial function of bounded degree**. Once such a piecewise–polynomial structure is established, we apply **Lemma 5 from Cheng et al. [1]** to obtain an upper bound on $\mathrm{Pdim}(\mathcal{U})$. Importantly, Lemma 5 yields **only an upper bound**, not an exact value.
> > >
> > > A similar approach is used in the branch-and-cut framework for mixed-integer optimization in [1], where only **upper bounds on $\mathrm{Pdim}(\mathcal{U})$** are derived. At present, we are not aware of a method that simultaneously provides a lower bound without leaving a significant gap between the two. Following your suggestion, we intend to pursue techniques for obtaining tighter (and potentially exact) bounds on **$\mathrm{Pdim}$** as part of our future theoretical work.
> > >
> > >
> > > [1] Sample Complexity of Algorithm Selection Using Neural Networks and Its Applications to Branch-and-Cut (NeurIPS 2024).

---

> ### Author Response · Authors · 2025-11-19
> **Response to Reviewer ntxk (Minor Comments)**
>
> **Minor comments.**
> All inconsistencies will be addressed in the new version. We reply to some of the minor questions here for clarification.
>
> 1. **Floor and indicator functions.**
> Both are used in our proofs. We forgot to define the ceil function.
>
> 2. **“h suddenly becomes a real vector.”**
> This is because the heuristic function is treated as a vector in $R^n$, where each state’s heuristic is one coordinate of the vector. Since we have a total strict order over states (Assumption 2), the heuristic for $v_i$ is $h(v_i)=h_i$ when $h$ lives in $R^n$.
>
> 3. **“Add ‘derived from S & Oki 2022’.”**
> Thank you for pointing this out. We use a different assumption and a different derivation in our proofs, but the resulting bound matches (Sakaue & Oki, 2022). We will add a clear explanation before the theorem.
>
> 4. **“Results in Valenzano ’14 hold regardless of admissibility.”**
> Yes, correct—this is what makes their results notable. We added a clear explanation of all heuristic assumptions used in their result.
>
> 5. **$\hat{H}$ undefined.**
> Thank you for noting this. $\hat{H}$ is the upper bound on the inadmissibility functions as a performance measure. We require a clear bound on every performance measure per Assumption 3. We have added the definition.
>
> 6. **$w_1 + \cdots + w_L$ is denoted by $U$, not $W$.**
> The notation $w_1 + \cdots + w_L = U$ is correct. Here, $U$ is the sum of layer widths, while $W$ is the total number of parameters (biases and matrices). For clarity:
>
> $$
> w_1 + \cdots + w_L = U
> $$
>
> and
>
> $$
> W = \sum_{l=1}^{L} \big( w_{l-1} w_l + w_l \big).
> $$
>
> 7. **Explanation of $\ell'$, $\ell$, $\hat{H}$, $H$, and $D$.**
> We clarify the differences. First, $H$ and $\hat{H}$ are upper bounds on the general performance measure and the inadmissibility functions, respectively. For example, $\hat{H}$ could be 20, meaning the maximum inadmissibility along an optimal path is at most 20. The symbols $\ell$ and $\ell'$ denote the number of heuristic classes considered by the network—the size of the last layer—when the dataset consists of states from the main graph $G$ or the PDB dataset $P$, respectively. It is also clear that $\ell' \leq \ell$, since the abstract state space is smaller than or equal to the main graph. For instance, in the 8-corner PDB, there are 12 distinct heuristic values, so $\ell'=12$. The term $D$ is the maximum heuristic value in the graph. With equal edge costs $c_0$ (Assumption 4), there are $\lceil D / c_0 \rceil$ distinct heuristic values. We provided this explanation in the proof of Theorem 5. In Rubik’s Cube, $c_0=1$, so $\ell=D$, but to apply the theorem to a broader class of graphs we distinguish $D$ from $\ell$ and $\ell'$.
>
> 8. **$|h|$ is never explicitly optimized.**
> Equation (4) formalizes the learning of neural network heuristics, highlighting model size as an explicit optimization objective so that practitioners do not end up with oversized models that are costly to query repeatedly during search. We optimized this parameter in Section I of the appendix, reducing it until heuristic quality remained unchanged. Also, all initial model sizes were chosen to be much smaller than recent approaches [5,7]. As seen in Table 2, model sizes increased by less than $2\times$, while dataset sizes increased by more than $10\times$.
>
> 9. **$Rep(I)$.**
> $Rep(I)$ must encompass all information required to find a path from the start state to the goal for instance $I$. We assume it contains the set $B$ of states encountered by running a search from the start state to the goal using $h^{*}$. This definition is mainly used in the appendix for proofs, but we also include it in the main paper, as $B$ appears in our bounds in Theorems 6 and 7.

---

> > ### Comment · Reviewer_ntxk · 2025-11-27
> > **RE: Response to Reviewer ntxk (Minor Comments)**
> >
> > Thanks for the clarifications.
> >
> > > $|h|$ is never explicitly optimized.
> >
> > This is not a critical comment, but it might be a good idea to add l1/l2 weight regularizations, as well as to cite papers on post-hoc weight pruning.

---

> > > ### Author Response · Authors · 2025-11-30
> > > **RE: model size and post-hoc weight pruning**
> > >
> > > Thank you for pointing this out. We missed citing the relevant papers on post-hoc weight pruning. We will include the appropriate citations in the updated version of the paper.
> > >
> > > We would also like to clarify that we are already using techniques related to this line of work. Specifically, we train all models in 32-bit precision, but perform inference in 16-bit precision to speed up the search. We also experimented with quantized (8-bit) inference; however, the accuracy drop was significant compared to the trained model, so we chose to report only the 16-bit inference results. We will add this explanation to the main paper as well.

---

> ### Author Response · Authors · 2025-11-19
> **Response to Reviewer ntxk (Questions)**
>
> ### Questions
>
> **Confirm the input format is PDB abstract state representation, factored.**
> Explained above. Yes, the input is the PDB abstract-state representation.
>
> **Confirm the target value is really $h^{*}$ and is obtained by solving $x$ optimally.**
> Yes. The target is the abstract-space $h^{*}$ stored in the PDB, obtained via optimal search.
>
> **Did you use the entire abstract states in the PDB without subsampling?**
> No. Please see the sampling strategy explained above.
>
> **I assumed $O(\sqrt{n})$ vs. $O(n)$ is what they implied. Am I correct?**
> Yes, you are correct.
>
> **Why is this a reduction in Theorem 3?**
> Please see Theorem 4.4 in [6], which provides a detailed proof.
>
> [1] https://github.com/nathansttt/hog2/tree/PDB-refactor
> [2] External Memory PDBs: Initial Results
> [3] Focal Loss for Dense Object Detection
> [4] Sample Complexity of Learning Heuristic Functions for Greedy Best-First and A* Search
> [5] Solving the Rubik’s Cube Without Human Knowledge
> [6] Worst-Case Solution Quality Analysis When Not Re-Expanding Nodes in Best-First Search
> [7] Self-Supervision is All You Need for Solving Rubik’s Cube
> [8] Compressed Pattern Databases

---

### Official Review · Reviewer_zAZ1 · 2025-11-01

**Soundness:** 2
**Presentation:** 2
**Contribution:** 2
**Rating:** 4
**Confidence:** 2

**Summary:**

This paper aims to learn admissible heuristics for A*, combining theoretical generalization analysis with practical training improvements. It incorporates a new loss function, called Cross-Entropy Admissibility (CEA), that enforces admissibility during training and analyzes the sample complexity of learning admissible heuristics. Experiments on the Rubik’s Cube show that CEA produces nearly admissible heuristics with very low inadmissibility rates and stronger guidance against compressed pattern databases (PDBs).

**Strengths:**

1. This paper introduces CEA as a principled loss to enforce heuristic admissibility is both theoretically sound and practically effective.
2. It provides theoretical analysis and derives tighter generalization bounds, incorporating neural networks as the heuristic with  generalization bounds analyzed primally on network size.
3. Experimental results demonstrates strong results on Rubik’s Cube domains, showing measurable gains over traditional PDBs.

**Weaknesses:**

1. The experiments results are limited to a single domain(the Rubik's Cube). The claims of the paper would be significantly stronger if supported by results on other classic planning domains.
2. The theoretical analysis relies on Assumption 4, while this holds for the Rubik's Cube but is not universal. How do the bounds change for graphs with variable edge costs?
3. The proposed method is evaluated on several 3 × 3 Rubik’s Cube pattern databases, where  the performance and data requirements for larger graphs remain unclear.

**Questions:**

Please refer to the weakness.

---

> ### Author Response · Authors · 2025-11-19
> **Response to Reviewer zAZ1**
>
> Dear Reviewer zAZ1,
>
> We appreciate your time on this review.
>
> ### Weaknesses
>
> **Experimental results are limited to a single domain (Rubik's Cube).**
> We agree that adding more puzzles would better demonstrate the applicability of our training framework. We chose Rubik’s Cube for two main reasons: first, there was previously no fully admissible learned heuristic for this domain—let alone a strong one—and second, its dynamics are particularly challenging, since actions induce highly nonlocal changes, making it harder for a model to capture relationships between consecutive states than in STP or ToH. For example, in the Sliding Tile Puzzle (STP), a single action changes only two tiles, and much of the representation remains the same between consecutive states. Regarding scalability and applicability, instead of adding more small PDBs from different domains, we used much larger PDBs within the same domain. Note that our largest PDB (7-edge) contains **510 million** states—exceeding one of the largest classification datasets [1].
>
> **Theoretical analysis relies on Assumption 4.**
> We use Assumption 4 in our theoretical results for two main reasons:
>
> 1. The first bounds on general graphs with a general performance measure are already established in [2]. Our goal was to improve these bounds for a specific class of graphs that satisfy Assumption 4.
> 2. To the best of our knowledge, PDB heuristics are available for graphs that satisfy Assumption 4. Since our goal was to use PDB heuristics to learn admissible heuristics, we incorporated this assumption into our theorems, which also tightened the bounds.
>
> Moreover, for upper bounds on the expected suboptimality, we do not use Assumption 4, and the results in that section apply to all graphs. Likewise, neither Theorem 6 nor Theorem 7 requires Assumption 4 to derive the bounds. We emphasize this assumption because our experiments focus on such graphs. The bounds in these theorems for a more general class of graphs can be obtained straightforwardly by using the bound $\mathcal{O}(n^2 \log n)$ instead of $\mathcal{O}(n \log n)$; we omitted these variants due to space constraints.
>
> **Performance and data requirements for larger graphs.**
> Could you please clarify what you mean by “larger graphs”? We have reported performance results for all graphs used, and the PDB dataset for each graph is sourced from [3]. The largest graph we evaluate is the 7-edge PDB containing **510 million** states. As future work, we plan to study an even larger graph: the 8-edge PDB with **5 billion** states (approximately 2.4 GB footprint), which is also available in [3]. For larger PDB datasets that provide stronger heuristics, please see [4].
>
> [1] Revisiting the Unreasonable Effectiveness of Data in the Deep Learning Era.
> [2] Sample Complexity of Learning Heuristic Functions for Greedy Best-First and A* Search.
> [3] https://github.com/nathansttt/hog2/tree/PDB-refactor
> [4] External Memory PDBs: Initial Results

---

> > ### Comment · Reviewer_zAZ1 · 2025-11-27
> > **Official Comment by Reviewer zAZ1**
> >
> > Thanks for the clarifications.
> > My main concerns have been addressed and I am raising my score accordingly.

---

> ### Comment · Reviewer_ntxk · 2025-11-27
> **A brief comment on review quality**
>
> As a reviewer who spend time and effort in reviewing this paper, I am frustrated by this low-quality review. Even if it was not produced by ChatGPT, this review should be treated as not passing the bar and this reviewer's submission should be rejected for not fulfilling the required duty.
>
> Disclaimer: I did not submit any paper to ICLR this year, and mine is not a simple attempt to play a zero-sum game.

---

### Official Review · Reviewer_bBv4 · 2025-11-03

**Soundness:** 3
**Presentation:** 3
**Contribution:** 3
**Rating:** 6
**Confidence:** 3

**Summary:**

This paper addresses the challenge of learning admissible heuristics for A* search. Many current deep learning approaches often do not consider admissibility but focus on predictive strength, losing theoretical guarantees on optimality. The authors propose new sample complexity bounds for learning such heuristics, that exploit graph structural properties and PDB abstraction. They also derive bounds for neural network that depend on the networks size rather than the problem size. From a practical point of view, the paper frames the heuristic learning as a constrained optimization problem and further reduce it to a classification task that uses a new loss function called cross-entropy admissibility. Empirical results show the method has low overestimation rate without sacrificing informativeness.

**Strengths:**

1. The formalization of heuristic learning as a constrained optimization and reduce it to a classification task guided by a new loss function is novel.
2. The tighten sample complexity bounds seem to be a good contribution.
3. The paper addresses a gap between ML-based heuristics and rigorous admissibility requirements for optimal search.
4. The paper is well-presented.

**Weaknesses:**

I wasn't able to check the theory thoroughly, the weaknesses listed below are mainly from empirical study.

1. Experiments are on Rubik’s Cube domain, it is a good benchmark but requires at least of couple more for the empirical results to be fully convincing. The practical value of CEA would be clearer if demonstrated in another admissible-heuristic domain (e.g., sliding-tile puzzle).
2. The paper didn’t describe how the hyper parameters \beta and \eta are tuned, if the author could provide a principled process, it would help reproducibility.
3. There is no runtime measurements in the experiments.

**Questions:**

1. Have you quantified A* search impact in terms of running time on your test problems?
2. Have you empirically evaluated your methods on other domains? How does it perform?

---

> ### Author Response · Authors · 2025-11-19
> **Response to Reviewer bBv4**
>
> Dear Reviewer bBv4,
>
> We appreciate your time on this review.
>
> ### Weaknesses
>
> **Experiments are on the Rubik’s Cube domain.**
> We agree that adding more puzzles would better demonstrate the applicability of our training framework. We chose Rubik’s Cube for two main reasons: first, there was previously no fully admissible learned heuristic for this domain—let alone a strong one—and second, its dynamics are particularly challenging, since actions induce highly nonlocal changes, making it harder for a model to capture relationships between consecutive states than in STP or ToH. For example, in the Sliding Tile Puzzle (STP), a single action changes only two tiles, and much of the representation remains the same between consecutive states. Regarding scalability and applicability, instead of adding more small PDBs from different domains, we used much larger PDBs within the same domain. Note that our largest PDB (7-edge) contains 510 million states—exceeding one of the largest classification datasets [1].
>
> **How are hyperparameters $\beta$ and $\eta$ tuned?**
> Please see Section J for full details. In summary, we do *not* tune these values. We start with an initial value for both hyperparameters and periodically reduce them if the desired admissibility is not achieved after a specific number of training epochs. Figure 4 in that section shows all values of $\eta$ and $\beta$ used during training. A simple way to automate this—used in the experiments of Table 5 in the appendix—is to halve both values over time if admissibility is not improving. Intuitively, reducing these two parameters indicates that the model cannot learn an admissible heuristic at the current strength, and we therefore encourage more underestimation to increase admissibility.
>
> **Runtime measurements.**
> All training-related hyperparameters are the same between the two loss functions to allow a fair comparison. The runtime follows the upper bounds provided in our theorems. We will release the complete log files and code upon acceptance.
>
> ### Questions
>
> **Running time of the A\* search.**
> This is a great question. Using learned heuristics for optimal search has two main components: (i) learning an admissible heuristic and (ii) designing efficient algorithms that exploit the learned heuristic. In this work, we study the theoretical and empirical aspects of learning an admissible heuristic. The search runtime—related to the second component—is studied in our accepted AAAI paper.
>
> **Evaluation on other domains.**
> We have not yet tested the approach on other domains. Since learning heuristic functions for the Rubik’s Cube has been a popular research topic, our goal here was to show that none of the learned heuristics to date (e.g., [2,3]) are admissible, and to derive the first strong admissible heuristics for this domain. Notably, our 8-corner model is $30\times$ smaller than DeepCube [2] while delivering an admissible heuristic.
>
> [1] Revisiting the Unreasonable Effectiveness of Data in the Deep Learning Era.
> [2] Solving the Rubik’s Cube Without Human Knowledge.
> [3] Self-Supervision is All You Need for Solving Rubik’s Cube.

---

> > ### Comment · Reviewer_bBv4 · 2025-11-27
> >
> > Thanks for the rebuttal. Can you summarize the runtime results from your AAAI paper? I can’t go look at it otherwise it violates double blind policy.

---

> ### Comment · Reviewer_ntxk · 2025-11-27
> **A brief comment on review quality**
>
> As a reviewer who spend time and effort in reviewing this paper, I am frustrated by this low-quality review.
> Even if it was not produced by ChatGPT, this review should be treated as not passing the bar and this reviewer's submission should be rejected for not fulfilling the required duty.
>
> Disclaimer: I did not submit any paper to ICLR this year, and mine is not a simple attempt to play a zero-sum game.

---

> ### Author Response · Authors · 2025-11-29
> **Summary of Runtime Performance**
>
> Dear Reviewer,
>
> Thank you for your response. Before summarizing the results, we would like to note that in the camera-ready version we will cite both the published version and the preprint version of the paper (including the additional experiments in the appendix), together with a clear explanation of the results.
>
> Now, let us summarize the empirical findings. In that paper, we argue that classical search algorithms are not efficient when used with ML-based heuristics. We compared node expansions and node generations under two settings:
>
> (1) our algorithm using ML heuristics
> (2) a classical search algorithm using PDB heuristics
>
> In our results, both the number of nodes generated and the number of nodes expanded were less than half of those for the PDB heuristics, even though both heuristics have the same memory footprint. This shows that the learned heuristic is **stronger** than the PDB heuristic.
>
> Regarding runtime, our algorithm running on four GPUs achieved performance very close to one of the state-of-the-art parallel search algorithms.

---

### Official Review · Reviewer_1RCQ · 2025-11-03

**Soundness:** 2
**Presentation:** 3
**Contribution:** 3
**Rating:** 6
**Confidence:** 2

**Summary:**

This paper tackles a core challenge at the intersection of machine learning and heuristic search: **how to learn powerful heuristic functions that preserve the optimality guarantees of algorithms like A***.

The authors make contributions on both theoretical and practical fronts:

*   **New Loss Function:** They propose a Cross-Entropy Admissibility (CEA) loss, a new training objective designed to enforce the admissibility constraint directly.

*   **Strong Empirical Performance:** Evaluated on the 3*3 Rubik's Cube, the approach demonstrates remarkable success, learning heuristics that are nearly perfectly admissible while providing stronger search guidance than traditional baselines.

*   **Theoretical Analysis:** The work provides a sample complexity analysis for learning heuristics. By leveraging problem structure and abstractions like Pattern Databases, the authors derive tighter generalization bounds for A*. They also establish a theoretical guarantee for learning goal-dependent heuristics.

In summary, this work introduces a framework for learning reliable heuristics, effectively bridging data-driven learning and classical search algorithms.

**Strengths:**

- **Strong Empirical Validation:** The work demonstrates compelling empirical results. On the challenging benchmark of the 3x3 Rubik's Cube, it achieves near-perfect admissibility while learning a heuristic stronger than classically compressed alternatives.
- **Noteworthy Theoretical Scope:** The paper provides a significant theoretical analysis of sample complexity. Its most original contribution is extending this framework to establish the first generalization guarantees for *goal-dependent* heuristics, moving beyond standard fixed-goal analyses.

**Weaknesses:**

- **Limited Empirical Scope:** The empirical evaluation is exclusively conducted on the 3*3 Rubik's Cube. The performance and effectiveness of the proposed CEA loss on other combinatorial puzzles or planning domains remain unknown.
- **Limitations of Theoretical Claim**: The claim that sample complexity is governed by network parameters and independent of graph size may not translate to practical advantages, as it relies on strong assumptions that may fail in real-world problems.
- **Unvalidated Theoretical Extension:** While establishing the first theoretical guarantees for goal-dependent heuristics is a notable contribution, this significant claim is unsupported by empirical evidence. The experiments only address a fixed-goal setting.

**Questions:**

- **Comparison with Standard Cross-Entropy Loss:** The paper clearly shows that the CEA loss achieves a much lower overestimation rate than the standard CE loss. However, beyond this crucial metric, did you compare the **resulting A\* search performance** (e.g., number of node expansions, wall-clock time) when using heuristics trained with these different losses?
- **Connection Between Theory and Experiment**: Could you share more details on how the theoretical insights guided the experimental design? For instance, did the theory influence the choice of neural network capacity or the amount of training data needed?

---

> ### Author Response · Authors · 2025-11-19
> **Response to Reviewer 1RCQ**
>
> Dear Reviewer 1RCQ,
>
> We appreciate your time on this review.
>
> ### Weaknesses
>
> **Limited Empirical Scope.**
> We agree that adding more puzzles would better demonstrate the applicability of our training framework. We chose Rubik’s Cube for two main reasons: first, there was previously no fully admissible learned heuristic for this domain—let alone a strong one—and second, its dynamics are particularly challenging, since actions induce highly nonlocal changes, making it harder for a model to capture relationships between consecutive states than in STP or ToH. For example, in the Sliding Tile Puzzle (STP), a single action changes only two tiles, the representation is a 2D image suitable for convolutions, and much of the representation remains the same between consecutive states. Regarding scalability and applicability, instead of adding more small PDBs from different domains, we used much larger PDBs within the same domain. Note that our largest PDB (7-edge) contains 510 million states—exceeding one of the largest classification datasets [1].
>
> **Limitations of Theoretical Claim.**
> We provide two upper bounds for learning heuristic functions with neural networks. In Theorem 5, the bound depends only on the graph size. The assumption is that the model is sufficiently expressive to realize all possible $\ell^n$ or $(\ell')^{m}$ heuristic combinations, depending on whether it is learned over the graph $G$ or the PDB dataset $P$. In Theorem 6, using Lemmas 3–6, we reduce this dependency, though it does not disappear entirely. The remaining dependency arises through the term $B$, the number of states encountered by running a search from the start state of a training instance to the goal using $h^*$. As the graph grows, $B$ naturally increases. In the worst case, $B$ can equal the graph size $|V| = n$, yielding the bound
> $$
> \mathrm{Pdim}(\mathcal{U}) = \mathcal{O}\left( LW \log\left( U + \ell \right) + W \log\left( \ell n (L+1) \right) \right).
> $$
>
> **Unvalidated Theoretical Extension to Goal-Dependent Heuristics.**
> We agree that including results for the goal-dependent setting would further support the theory. The challenge is the lack of available datasets for this case. For instance, our smallest PDB has 42M states; the corresponding goal-dependent dataset would have size $1.76\times 10^{15}$. No such dataset exists in the literature, so we were unable to provide results.
>
> ### Questions
>
> **Comparison with Standard Cross-Entropy Loss.**
> This work studies theoretical and empirical results for learning an admissible heuristic that guarantees the optimal solution for every state. To date, no such heuristic has been learned for Rubik’s Cube. Our method yields fewer node expansions than the PDB heuristic with the same memory footprint because it achieves a larger average heuristic while maintaining a very similar admissibility rate. It also outperforms CE, as CE assigns inaccurate values due to its higher inadmissibility. Note that the average heuristic is a meaningful proxy for heuristic strength when the heuristic is admissible or near-admissible. Please see the distributions for both loss functions in Figures 6, 7, and 8. Wall-clock search time depends on the search algorithm; we study the design of efficient search algorithms in our recently accepted AAAI paper.
>
> **Connection Between Theory and Experiment.**
> This is a great question. The theory and empirical results are linked via the PDB datasets. PDBs reduce the sample-complexity bound (number of training instances) and enable learning an admissible heuristic over the entire state space. The model architecture for all experiments (a ReLU network) follows from Theorem 6. The training duration is chosen to reach the desired admissibility level; afterward, we verify that this aligns with the theorem’s bound.
>
> [1] Revisiting the Unreasonable Effectiveness of Data in the Deep Learning Era.

---

> > ### Comment · Reviewer_1RCQ · 2025-11-27
> >
> > Thanks for the comments. After a careful reading of the clarification by authors and the reviews from other reviewers, I would like to maintain my rating.

---

### Official Review · Reviewer_BkY5 · 2025-11-03

**Soundness:** 4
**Presentation:** 4
**Contribution:** 3
**Rating:** 8
**Confidence:** 3

**Summary:**

The paper presents a method, called cross-entropy admissibility (CEA), to learn heuristic functions which, in the context of pathfinding on a weighted directed graph, map states to an estimate of the cost-to-go to a closest goal state. CEA focuses on learning heuristic functions that are admissible, meaning they never overestimate the cost of a shortest path. CEA formulates heuristic learning as a classification task, where each possible heuristic value is treated like a class, and more emphases is put on penalizing overestimation. The paper focuses on the Rubik’s cube, where all transition costs are one and, therefore, all possible true cost-to-go values are integers. Results with neural networks on learning heuristic values from PDBs show that the proposed method learns heuristic values that have significantly fewer underestimations than regular cross entropy (CE) while having a slightly less or similar average heuristic value.

**Strengths:**

As machine learning continues to be used to learn heuristic functions, admissibility guarantees become more important. The paper contains important theoretical analyses and proposed practical approaches for learning heuristic functions that are largely admissible.

When compared to compressed PDBs, both the CEA and CE have higher average heuristic values while maintaining similar compression rates. However, CE, has overestimation rates on the order of 1o^-3 to 10^-2. On the other hand, CEA has compression rates on the order of 10^-7 to 10^-5. The average heuristic value of CEA when compared to CE ranges from slightly lower to having the same value.

**Weaknesses:**

The paper discusses the benefits of admissibility, but it is not clear at what the cost of prioritizing admissibility on problem solving ability. If a neural network uses more of its representational capacity to prevent overfitting, then it may stand to reason that it may perform relatively poorly on higher cost-to-go values. This could make finding paths more time consuming in practice due to the lower overall heuristic value. Table 2 indicates this could be the case due to CEA having a slightly lower average heuristic value compared to CE.

**Questions:**

Does the theory presented in the paper say anything about drawbacks from prioritizing admissibility?

For Table 2, has using mean-squared error been tried?

---

> ### Author Response · Authors · 2025-11-19
> **Response to Reviewer BkY5**
>
> Dear Reviewer BkY5,
>
> We appreciate your time on this review.
>
> ### Weaknesses
>
> **Cost of prioritizing admissibility on problem-solving ability.**
> This is a great question. The optimality of the solution is not the same as the efficiency or runtime of the search. An admissible heuristic guarantees an optimal solution, but it may take longer to find than an inadmissible heuristic that yields a suboptimal solution for the same state. In this work, we focus on guaranteeing optimality for learned heuristic functions. These can also be used for finding solutions with bounded suboptimality.
>
> Regarding the comparison of CE’s average heuristic with CEA, a larger average heuristic does *not* necessarily imply a stronger heuristic that yields fewer node expansions. Average heuristic value is a valid metric when both heuristics are admissible or near-admissible. For example, a heuristic for the 8-corner PDB that returns a value of 10 for all states has an average of 10, yet performs very poorly in search, causing many node expansions. In our case, CE has a high inadmissibility rate; these errors are misleading and lead to substantially more node generations (higher runtime and lower efficiency). Please see the heuristic distributions for both loss functions in Figures 6, 7, and 8.
>
> ### Questions
>
> **Theory reflecting admissibility.**
> In our theoretical results, we aim to bound the number of training samples needed to guarantee generalization of A* performance to unseen data. By “generalization,” we mean that the strong performance observed on the training set carries over to the test set (unseen states). Since our training set consists of admissible heuristics (PDBs), “good performance” is defined as the ability to find optimal solutions. Thus, our goal is to show how many training samples are required to guarantee optimal solutions on unseen data. Because “good performance” is defined as optimality in solution length, admissibility is already reflected in the setup and does not change the bounds. It is also worth noting that an admissible heuristic enables the use of a broad range of search algorithms that guarantee bounded-suboptimal solutions efficiently [1].
>
> **Mean-squared error.**
> This is a great question. Our first loss to try was mean-squared error, which performed poorly in terms of admissibility, even on smaller PDBs. The issue stems from regression’s objective of predicting a real-valued output for every state. Even when many states share the same true heuristic, the model’s continuous predictions do not match exactly, so admissibility is frequently violated. For example, suppose two states $s_1$ and $s_2$ satisfy $h^\*(s_1) = h^\*(s_2) = 7$. The model instead predicts $h(s_1) = 7.0931$ and $h(s_2) = 6.99901$. Then $h(s_1)$ is inadmissible despite being “close.” We attempted to fix this by subtracting a constant from all predictions to handle such edge cases, but this reduced the average heuristic value, resulting in a weaker heuristic.
>
>
>
>
> [1] Using Alternative Suboptimality Bounds in Heuristic Search.

---

> ### Comment · Reviewer_ntxk · 2025-11-27
> **A brief comment of review quality**
>
> As a reviewer who spend time and effort in reviewing this paper, I am frustrated by this low-quality review. Even if it was not produced by ChatGPT, this review should be treated as not passing the bar and this reviewer's submission should be rejected for not fulfilling the required duty.
>
> Disclaimer: I did not submit any paper to ICLR this year, and mine is not a simple attempt to play a zero-sum game.

---

### Official Review · Reviewer_KNcH · 2025-11-05

**Soundness:** 3
**Presentation:** 3
**Contribution:** 3
**Rating:** 6
**Confidence:** 2

**Summary:**

This paper introduces a new loss function, Cross-Entropy Admissibility (CEA), for learning A* heuristics that are both strong and nearly admissible. The authors show that CEA explicitly favors underestimation while still concentrating probability on the true heuristic value, yielding more effective heuristics than compressed pattern databases in domains such as the Rubik’s Cube.

On the theoretical side, the paper provides generalization guarantees for learned (including goal-dependent) heuristics in A*, deriving improved sample complexity bounds by analyzing inadmissibility along optimal paths. Overall, the work makes both practical and theoretical contributions to learning heuristics for search, bridging a gap between empirical performance and provable guarantees.

**Strengths:**

Novel loss function tailored to A*: The proposed CEA loss directly encodes admissibility preferences, addressing a key limitation of prior heuristic-learning approaches that treat over- and under-estimation symmetrically.

Strong empirical performance: Experiments (e.g., on Rubik’s Cube) demonstrate that the learned heuristics are nearly admissible while significantly outperforming compressed PDB heuristics of equal memory footprint, showing both practical effectiveness and scalability.

Theoretical contribution with improved sample complexity bounds.

**Weaknesses:**

Fairness of experimental comparison could be improved: While the comparison with compressed PDBs under equal memory budget is meaningful, it overlooks other relevant dimensions such as training cost, inference time, and tuning effort. In particular, CEA models were tuned, whereas the CE baseline shares hyperparameters without comparable tuning, which may bias results in favor of CEA.

CEA introduces additional tuning complexity

Limited domain diversity in evaluation: Experiments focus primarily on the Rubik’s Cube domain. It remains unclear how well the method generalizes to other search domains with different structure, branching factors, or cost models.

**Questions:**

The authors mention that the CE and CEA models use the same parameters. Could the authors clarify how these parameters were chosen? For a fair comparison, it may be helpful to tune both loss functions under the same hyperparameter-search budget. It would also be valuable to report the tuning ranges, the best-found values.

While comparing the NN heuristic with a compressed PDB under an equal memory budget is a meaningful and reasonable fairness criterion, may I ask whether memory alone is sufficient to capture the trade-offs between these approaches? Would it be possible to also comment on the differences in training cost and inference time, as these aspects are often relevant for practical deployment?

---

> ### Author Response · Authors · 2025-11-19
> **Response to Reviewer KNcH**
>
> Dear Reviewer KNcH,
>
> We appreciate your time on this review.
>
> ### Weaknesses
>
> **Fairness of experimental comparison.**
> We view this concern as having two aspects and address each separately.
>
> **CE vs. CEA:**
> All learned models—trained with either CEA or CE loss—underwent the same amount of training (number of epochs). By “the same hyperparameters” for both losses, we mean all training-related hyperparameters (model size, optimization algorithm (Adam), learning rate, and batch size) were tuned equivalently for both approaches. The only additional parameters are $\beta$ and $\eta$, introduced by CEA. However, these parameters do not require manual tuning. Training with CEA starts from an initial setting aimed at learning a perfectly admissible heuristic; then, periodically during training, if the desired admissibility rate is not achieved after a set number of epochs, we reduce both hyperparameters (e.g., halve them) to trade perfection for higher admissibility. This schedule is used for the models in Table 5 (appendix). The values of $\beta$ and $\eta$ for all models used in the Table 2 experiments are shown in Figure 4.
>
> **PDB heuristics vs. CEA-learned heuristics:**
> We agree that the comparison should consider more than memory footprint. Our goal was to demonstrate that deep learning can yield stronger heuristics than traditional compression techniques in classical planning. So in future, we can use deep learning to compress much larger PDBs (e.g., the 12-edge Rubik’s Cube PDB, ~5.8 trillion states) rather than relying on classical compression. Regarding training cost, note that constructing strong PDB heuristics for arbitrary domains is itself nontrivial and requires design effort. Inference time is critical, which is why our optimization objective includes minimizing model size. End-to-end search runtime further depends on an algorithm engineered to exploit NN heuristics efficiently; importantly, NN heuristic lookups can be batched, enabling multiple states to be evaluated in parallel. We study the design of efficient search algorithms in our recently accepted AAAI paper.
>
> **Limited domain diversity in evaluation.**
> This is a fair point. Classical planning includes other domains such as the Sliding-Tile Puzzle (STP) and Tower of Hanoi (ToH), in addition to Rubik’s Cube. We focused on Rubik’s Cube because its dynamics are particularly challenging: actions induce highly nonlocal changes, making it harder for a model to capture relationships between consecutive states than in STP or ToH. Rather than adding more domains, we evaluated scalability by increasing PDB size; in our largest PDB (7-edge PDB), there are 510 million states—exceeding one of the largest classification datasets [1].
>
> ### Questions
>
> **Parameters used by CEA and CE.**
> We agree that equal effort should be invested in obtaining results for both loss functions. We guarantee that all training-related parameters—especially the number of epochs—are matched across approaches. CE tends to underperform because it treats overestimation and underestimation symmetrically, pushing the model to learn a perfect heuristic for every state without emphasizing admissibility, which leads to getting stuck in a local optimum. The best hyperparameters for all models appear in Table 4 (appendix), and the best $\beta$ and $\eta$ values for the Table 2 models are shown in Figure 4.
>
> **Training cost and inference time for practical deployment of CEA.**
> This is an important question. The runtime of a search algorithm using a learned heuristic depends on more than the heuristic’s model size, though model size is the primary driver of inference latency on the heuristic side. To show how optimized our model sizes are, note that our 8-corner model is $30\times$ smaller than DeepCube [2] while delivering an admissible heuristic. To optimize end-to-end runtime, the search algorithm must also be designed for NN heuristics. Note that comparing raw inference time is tricky because NN heuristics benefit from batched evaluations, which can substantially reduce effective inference time. As for training cost, it is a one-time pre-deployment expense; the trained heuristic can then be reused across every problem instance in the same domain to produce optimal solutions. We will release code and training logs to document training time and other details upon acceptance.
>
> [1] Revisiting the Unreasonable Effectiveness of Data in the Deep Learning Era.
> [2] Solving the Rubik’s Cube Without Human Knowledge.

---

### Author Response · Authors · 2025-11-19
**New Version of the paper is uploaded.**

### Dear Reviewers

We appreciate your time and effort in providing reviews for our paper. We have uploaded a new version that addresses the concerns you raised. Below is a summary of the main edits:

1. Added a more detailed description of the training procedure.
2. Provided additional explanations regarding PDBs and delta-PDBs.
3. Included further elaboration on the theoretical results related to neural networks.

---

### Author Response · Authors · 2025-11-30
**Summary of Revisions and Global Response to Reviewers and AC**

We thank all reviewers for their time and thoughtful feedback. During the discussion period, we revised the paper based on your comments and also refined several aspects of the work’s future directions.

## **Final Version of the Paper**

The final version of the paper includes the following updates compared to the submitted version:

1. Added a more detailed description of the training procedure.
2. Provided additional explanations regarding PDBs and delta-PDBs.
3. Included further elaboration on the theoretical results involving neural networks.
4. Clarified the techniques used to optimize model size and cited relevant post-hoc weight-pruning methods.
5. Added a new experiment examining how the generalization error behaves as the number of training instances increases, providing empirical support for the sample-complexity analysis.

## **Main Contributions of the Paper**

For clarity, we restate the main contributions of the work:

1. **Sample-complexity analysis.**
   We analyze the sample complexity of learning heuristic functions from data and introduce a new upper bound on the expected suboptimality of A* with reopenings. We then derive tighter generalization bounds by leveraging the structural properties of the graphs of interest. Incorporating neural networks as the heuristic model yields generalization bounds that depend primarily on network size rather than graph size. Notably, we provide the first generalization bounds for goal-dependent heuristics. Across all theoretical results, we show that leveraging PDBs—rather than sampling from the original graph—produces improved bounds.

2. **Learning admissible heuristics.**
   We formulate the optimization problem for learning admissible heuristics for classical search and propose a new loss function, **Cross-Entropy Admissibility (CEA)**, together with a training framework that enforces all required constraints. Evaluated on several $3 \times 3$ Rubik’s Cube PDBs, CEA produces near-admissible heuristics, achieving inadmissibility rates around $10^{-6}$, significantly outperforming heuristics trained with standard Cross-Entropy (CE). The strength of the learned heuristics surpasses that of classical compressed PDBs of equal size; for the 8-corner PDB, CEA even learns the admissible PDB perfectly.

3. **Empirical validation of generalization behavior.**
   We empirically demonstrate that the generalization error decreases as the number of training instances increases, following the decay predicted by our sample-complexity analysis.

We hope that the revisions made during the discussion period have addressed the concerns raised by the reviewers, and we appreciate the opportunity for continued discussion. Before the scores were reverted, **all seven reviewers had assigned scores of 6 or higher**, thanks in particular to reviewers **zAZ1** and **BWMp**, who **raised their scores from 4 to 6** after their concerns were addressed during the rebuttal. Based on the email received from the ICLR Program Chairs, we cannot confirm whether these score increases occurred *before* the leak; however, we can state with confidence that **the score changes were made solely in response to the rebuttal and the clarifications we provided**.

Sincerely,
The Authors

---

### Meta-Review · Area_Chair_gHcf · 2026-01-03

**Summary:**

I will list the most important comments that the reviewers noted during the review process:
1) Fairness and completeness of experimental comparison could be improved, for example the experiments only address a fixed-goal setting.
2) CEA introduces additional tuning complexity.
3) Limited domain diversity in evaluation
4) The theoretical analysis relies on Assumption 4, while this holds for the Rubik's Cube but is not universal.
5) Disconnect between theoretical results vs experiments, for example the paper does not empirically verify Theorem 4.
6) The paper didn’t describe how the hyper parameters \beta and \eta are tuned.
7) There is no runtime measurements in the experiments.
8) It lacks a clear motivation why CAE specific function was chosen.

**Reviewer Concerns:**

Some of the comments mentioned above were correctly addressed by the authors, namely:
1) The theoretical analysis relies on Assumption 4, while this holds for the Rubik's Cube but is not universal.
2) The paper didn’t describe how the hyper parameters \beta and \eta are tuned.
3) There is no runtime measurements in the experiments.
4) It lacks a clear motivation why CAE specific function was chosen.

The authors agreed with some of the shortcomings or they remained unaddressed:
1) Fairness and completeness of experimental comparison could be improved, for example the experiments only address a fixed-goal setting.
2) CEA introduces additional tuning complexity.
3) Limited domain diversity in evaluation.
4) Disconnect between theoretical results vs experiments, for example the paper does not empirically verify Theorem 4.

**Reviewer Scores:**

1) Reviewer KNcH (score 6) would most likely have left his initial score.
2) Reviewer BkY5 (score 8) would most likely have left his initial score.
3) Reviewer 1RCQ (score 6) explicitly confirmed that he would leave his initial score.
4) Reviewer bBv4 (score 6) would most likely have left his initial score.
5) Reviewer zAZ1 (score 4) confirmed explicitly that he would raise his initial score.
6) Reviewer ntxk (score 6) could raise his score.
7) Reviewer BWMp (score 4) confirmed explicitly that he would raise his initial score.

---

### Decision · Program_Chairs · 2026-01-26

Accept (Poster)